# Development and Validation of Monte Carlo Methods for Converay: A Proof-of-Concept Study

**DOI:** 10.3390/cancers17071189

**Published:** 2025-03-31

**Authors:** Rodolfo Figueroa, Francisco Malano, Alejandro Cuadra, Jaime Guarda, Jorge Leiva, Fernando Leyton, Adlin López, Claudio Solé, Mauro Valente

**Affiliations:** 1Centro de Excelencia de Física e Ingeniería en Salud (CFIS), Universidad de La Frontera, Temuco 4811230, Chile; francisco.malano@ufrontera.cl (F.M.); jaime.guarda@ufrontera.cl (J.G.); jorge.leiva@ufrontera.cl (J.L.); leyton.fernando@gmail.com (F.L.); adlin.lopez@ufrontera.cl (A.L.); 2Departamento de Ciencias Físicas, Universidad de La Frontera, Temuco 4811230, Chile; 3Clínica IRAM, Santiago 7630370, Chile; alejandro.cuadra@iram.cl (A.C.); claudio.solep@iram.cl (C.S.); 4Facultad de Medicina, Universidad Diego Portales, Santiago 8370067, Chile; 5Instituto de Física E. Gaviola (IFEG), CONICET & Facultad de Matemática, Astronomía, Física y Computación (FAMAF), Universidad Nacional de Córdoba, Córdoba 5000, Argentina

**Keywords:** convergent beam radiotherapy, CONVERAY system, Monte Carlo simulation

## Abstract

The CONVERAY project introduces an innovative teletherapy system featuring a convergent X-ray beam, designed to achieve highly conformal dose distributions. This proof-of-concept study uses Monte Carlo techniques to model passive and active CONVERAY device components to investigate its preliminary dosimetry performance on realistic patient-specific intracranial and chest irradiation conditions. Simulation results demonstrate the system’s potential to deliver high dose conformation to the studied complex clinical targets, warranting further development and validation.

## 1. Introduction

In the past decades, technological improvements in diagnostic imaging, treatment planning and delivery enabled more accurate treatment of diseased tissue avoiding side-effects on healthy tissues. The radiotherapy technology particularly rapidly evolved, thus making available many innovative radiotherapy treatment options [1,2,3]. Although modern technological advances, such as hadrontherapy, often reduce toxicity and improve patient outcomes [4], they are also associated with increased financial costs, thus producing substantial differences among high- and low-income countries [5]. Recently proposed novel teletherapy devices based on convergent X-ray beams [6,7] have proven an ability to attain a highly concentrated dose delivery to the target, thus providing alternatives to highly conformed dose distributions by means of a photon spread out peak deposition because of the inherent converging properties.

The CONVERAY project proposes an innovative system for teletherapy based on the application of a convergent photon (X-ray) beam. The CONVERAY device is [7,8] designed to adapt to the head of current linear accelerators and the converging beam photon effect is achieved by producing a bremsstrahlung, thus treatment volumes can be placed in the focal spot where high fluence rate is achieved due to the beam convergence [7,8,9].

Conventional radiotherapy systems rely on diverging radiation beams, which require collimation to achieve targeted dose concentrations. In contrast, converging beams offer an alternative approach to achieving highly targeted dose conformation. Unlike divergent radiotherapy approaches, which require overlapping fields from multiple angles to achieve high dose concentrations, the CONVERAY system inherently attains this goal through its photon spread-out peak. This proof-of-concept study focuses on the methodological development and technical validation of the CONVERAY system, demonstrating its potential for improved dose conformity through computational simulations.

Monte Carlo (MC) methods largely proved to be one of the most accurate tools to solve problems in the field of medical physics [10,11,12]. Actually, the use of MC methods in radiotherapy dosimetry increased almost exponentially in the last decades mainly due to the computing power growth, becoming nowadays a common tool for reference and treatment planning dosimetry calculations [13,14]. All relevant dosimetry quantities, such as fluence, Kerma, and dose can be accurately calculated by MC simulations, even considering complex source/mass configurations [13,15]. Hence, assessing the dosimetry performance of the novel CONVERAY system by means of MC simulations appears to be the most suitable preliminary approach. Although from a general perspective convergent photon beam radiotherapy, such as CONVERAY, may offer potential advantages due to its inherent capability to achieve high dose concentrations, detailed investigations are still necessary to depict its reliability by fully characterizing the relevant components and operation process.

This work summarizes the fundamental principles for describing converging beams’ dosimetry performance, along with the application to the CONVERAY system. The step-by-step radiation fluence and interaction processes inside the CONVERAY device, as coupled to a conventional linear accelerator head, are studied for different CONVERAY configurations, obtaining in all cases complete phase space characterizations that are further used for dosimetry purposes. Finally, the preliminary dosimetry performance is reported for representative intracranial and chest irradiations, evaluating the expected capacity to attain high dose concentration within the targets. The main outcomes obtained for the preliminary dosimetry performance of the CONVERAY system involve the following: (*i*) detailed physical/radiological characterization of the different setups by means of associated phase spaces, (*ii*) in-phantom dosimetry outputs assessment, and (*iii*) patient-specific dosimetry calculation. The CONVERAY system was confirmed to be capable of attaining high dose conformation to complex targets, supporting further investigations, both at computational and prototype levels aimed at evaluating its definitive dosimetry performance.

## 2. Materials and Methods

Although the main concept and features behind the CONVERAY system have been previously described in detail [7,8] this section provides information on the methodologies and specific processes carried out to achieve the results herein reported.

### 2.1. Analytic Approach to the CONVERAY System

A basic sketch depicting the convergent beam is reported in Figure 1. The in-depth on-axis dose distribution (*D*(*z*)), as approached by the primary component, can be estimated according to Expression (1):(1)$$D(z)=D_{0}{\left(\frac{A(z=0)}{A(z)}\right)}^{2}e^{-\int_{0}^{E}\;\int_{o}^{z}\;\mu \left(E,z\right)\;\rho (z)\;dE\prime \;ds}$$
where *s* = *h sec* (*φ*), *μ* (*E, z*) is the mass attenuation coefficient at photon energy *E* and depth *z*, *A*(*z*) indicates the transversal area at depth *z*, while $\dot{D_{0}}$ and $\dot{D_{F}}$ are the dose rate at the incident surface and the focal spot, respectively, according to the CONVERAY primary beam component [7,8].

Therefore, the peaked nature of the resulting in-depth dose distribution allows a suitable longitudinal scan to produce in-depth dose profiles analogous to the spread-out Bragg peak produced by protons/hadrons. This effect, called photon spread-out peak (PSOP), may be useful for highlighting the potentiality of convergent beams to attain in-depth highly conformed dose distributions, as sketched in Figure 2.

Dose rate assessment by means of Expression (1) can be straightforward in terms of fluence rate. Thereby, the absorbed dose rate per electron current is reported herein, which will be particularly helpful for linking analytic and Monte Carlo calculations with linac operation parameters.

### 2.2. Monte Carlo Modeling of the CONVERAY System

Different MC main codes have proven to provide an accurate description of radiation transport and dosimetry effects for a wide range of medical physics applications, even considering fully realistic clinical conditions [11,12,13,14]. This subsection summarizes the main issues related to the implemented MC approach used to characterize the CONVERAY system, and particularly its dosimetry performance.

For the purposes of this proof-of-concept study, different subroutines have been adapted from two MC main codes. The integral CONVERAY system has been modeled by means of the FLUKA code [15] in order to achieve the overall performance for the different configurations, along with the associated phase space associated to each CONVERAY setup, which can be further used as input for independent simulations to characterize the corresponding dosimetry performance. Figure 3 reports a general overview of the CONVERAY device as defined and visualized by the FLAIR graphical interface for FLUKA, showing the main components and beam production stages: (1) primary/incident electron beam flowing in the +z direction, (2) virtual detectors (no effects no radiation transport) to characterize the particle fluence along the path, (3) the CONVERAY enter sheet (3 mm cylindrical collimator), (4) the first deflection magnet characterizing the magnetic field spatial distribution by a suitable 3D voxelization (uniform magnetic field intensity within each millimetric-sized voxel), (5) the CONVERAY “head” made of shielding materials, the “guided” electron beam enters the piece through the 10 mm diameter cylindrical canal to reach (6) the second deflection magnet also defined by 3D millimetric-size uniform magnetic field voxels, (7) the target and the filter that can be made of different combinations of materials, commonly a 1.15 mm thick W disc for the target, and (8) the emerging (photon) beam W collimator, whose diameter and length can be adjusted for the different collimation configurations. Typical representative values are as follows: 10–30 mm diameter and 80–120 mm length, and, finally, (9) the irradiated phantom/patient.

Then, the phase spaces obtained as output from the FLUKA simulations have been used as input for PENELOPE-based [16] simulations, along with adapted subroutines to perform 3D absorbed dose distribution on CT-based voxel-level patient-specific clinical cases. Additionally, the 3D-Slicer software version 5.8.1 has been used to assist with the RT-STRUCT data as well as further dose–volume histogram calculation.

Effects due to the magnetic fields have been accounted by the dedicated adaptation of the *magfld.f* FLUKA subroutine, introducing a suitable voxel-level definition of the magnetic field spatial distribution in accordance to experimental characterization of the deflection magnets using calibrated Gaussmeter and micrometric positioning systems [17,18]. For instance, Figure 4 reports the magnetic field central plane for one of the deflection magnets by illustrating the magnetic field strength in the central plane of the first deflection magnet, using corresponding experimental data that further serve as input for an accurate simulation of the electron beam along the intended path. Therefore, the uniformity of the magnetic field, as depicted in Figure 4, is critical for ensuring precise trajectory control of the electron beam within the CONVERAY system. This uniformity minimizes beam divergence and energy loss, which are essential for achieving the high fluence and dose conformity required for the system’s performance.

Then, the phase spaces obtained as output from the FLUKA simulations have been used.

#### 2.2.1. Phase Space for the CONVERAY Prototype

The Boltzmann radiation transport formalism can be stated in terms of the expected radiation density from collisions or a source in a unit volume of phase space depending on all relevant radiation field (phase state) coordinates, commonly a 7D space: vectorial position *r*, kinetic energy *E*, vectorial direction *Ω*, and time *t*. Monte Carlo simulation codes track the evolution of each particle path in the shower, thus position, energy, and direction are always available during the particle tracking. Thereby, some Monte Carlo codes allow the users to adapt/develop a dedicated subroutine to obtain the phase state, saving the information (phase state variables) of each particle in an output file that can be further used as a radiation source.

For the purposes of the present work, the *mgdraw.f* subroutine of the FLUKA MC code has been adapted to produce the output file containing the phase space of all particles emerging from the CONVERAY device. It is worth mentioning that the final phase space used to initiate further simulations corresponds to a complete 360° turn of the dynamic operation mode.

The information of the steady-state phase state variables has been organized in one row per emerging particle and columns report: *K-PAR* (particle type), *E* [MeV], *x* [cm], *y* [cm], *z* [cm], *u*, *v*, and *w*; where *r* = (*x*, *y*, *z*) and *u*, *v*, and *w* represent the direction cosines with respect to the positive *x*-, *y*-, and *z*-axis, respectively.

#### 2.2.2. In-Phantom Dosimetry and Dose Concentration Assessment for CONVERAY

The dosimetry performance of the CONVERAY prototype has been studied for realistic patient-specific cases representative of stereotactic radiosurgery (SRS) and stereotactic body radiotherapy (SBRT) aimed at evaluating the capability to concentrate an absorbed dose within a complex target for intracranial and thoracic irradiations. In both cases, voxelized geometries were used for the simulations, which have been inferred from anonymized CT *dicom* images of the selected clinical cases.

For the purposes of this work, CT images have been segmented into different regions using the “Segment Editor” module of the open access *3DSlicer* software. The intracranial irradiation implemented image segmentation by appropriate thresholding of the Hounsfield units (HU) of each voxel into regions corresponding to soft tissue, compact bone, and air. On the other hand, for thoraxic irradiation, a thresholding of the HU, has been previously performed in regions corresponding to soft tissues, compact bone, and air, and then the volumes corresponding to the lungs have been segmented using as reference the demarcation of the same made by the radio-oncology specialist, thus leaving the image segmented into four different regions, in complete accordance with the anatomic regions depicted in the *RT-STRUCT dicom* files, which could be successfully managed by the *SlicerRT* module. Then, the segmented images were converted to a file format that can be read by PENELOPE’s *PenEasy* package.

The primary radiation source used to initiate the beam used in the simulations with PENELOPE was defined from the phase spaces obtained with the FLUKA code, according to the procedure described in Section 2.2.1. In this framework, PENELOPE-based simulations made it possible to obtain the spatial dose distributions per primary particle at voxel level, as defined by the CT grid.

Finally, the obtained dose distributions have been read and processed using the “*SlicerRT*” module, which is provided as an extension of the *3DSlicer* software. Using this module, the isodose curves and the dose–volume histograms (DVH) of the spatial dose distributions have been evaluated.

### 2.3. Uncertainties of the CONVERAY Monte Carlo Modeling

While Monte Carlo simulations have proven to be a reliable and accurate method for describing radiation transport in complex systems, it is essential to acknowledge that computational modeling outputs inherently carry uncertainties. These uncertainties arise from various sources, including systematic and statistical factors, which can impact the accuracy and reliability of the results.

To quantitatively evaluate the impact of system variables on overall accuracy, a dedicated approach has been implemented. This strategy distinguishes between two primary types of uncertainties: (*i*) systematic uncertainties: these arise from modeling features influenced by pre-established conditions, including particle tracking algorithms, radiation-matter interaction models, observables’ calculation theory, geometry management algorithms (e.g., interface crossing, magnetic field description), virtual sketch representation and key simulation parameters (e.g., absorption/cutoff energy, simulation schemes), and (*ii*) statistical uncertainties: these stem from the inherent stochastic nature of the Monte Carlo method, as well as internal processes such as variance reduction techniques.

Table 1 outlines the approach used to evaluate the accuracy of the MC simulation process, considering the impact of various relevant parameters. The simulations initiate with the definition of the primary particles’ initial state. In this study, Gaussian probability distributions have been employed to describe the primary particles’ properties, striking a balance between realism and practicality. Specifically, the mean value and full width at half maximum (FWHM) of the electrons’ initial kinetic energy (*<E_el_>*, *FWHM* (*E_el_*)) and the beam propagation direction, characterized by the angle α between the particle propagation vector and the optical (idealized) axis, have been identified as key properties to investigate throughout the simulation process to ensure accurate dosimetry output. Note that modifying initial parameters, such as kinetic energy, necessitates a corresponding adjustment to the setup configuration to ensure proper beam guidance. To address this, minor tilts are applied to the deflecting magnets.

According to the outcomes from the preliminary assessment regarding the influence of the different parameters resumed in Table 1, proper knowledge about the primary beam energy spread (*FWHM* (*E_el_*)), as well as its corresponding propagation direction spread (*FWHM_α_*), notably impacts the further stages of the convergent photon beam production. Conversely, less instability corresponds to the uncertainties in the initial energy mean value (*<E_el_>*) and the initial beam propagation direction (*α_beam_*). Despite the outcomes resumed in Table 1, specific efforts have been invested to characterize the impact of uncertainties in other relevant parameters on the main final dosimetry output. To this aim, the absorbed dose per primary history (*shower*) is calculated within a 1 cm radius sphere concentric with the CONVERAY focal spot, and differences (*ΔD*) are obtained as percentage variations with respect to the reference case, as depicted in Table 2. Moreover, the fraction of the required computation time in terms of the reference case (*f_t_*) is also provided as a practical feature to be accounted for.

In summary, the findings on the effects of detailed and condensed simulation approaches are consistent with expected trends [17,18]. To strike a balance between accuracy and computational efficiency, we employed the following settings: *E_Abs_* = (1 × 10^4^, 1 × 10^4^) eV and *N_Tot_* = 2 × 10^9^. Additionally, a mixed simulation scheme is also implemented using PENELOPE and FLUKA codes, which combines detailed simulation for photons and charged particles with kinetic energies below 1 MeV with condensed approximations for energies above 1 MeV.

## 3. Results

This section presents the key findings from the development of the CONVERAY system, encompassing the following: (*i*) primary fluence and photon beam production, characterized by phase space analysis, and (*ii*) dosimetry performance of the CONVERAY system in both phantom and patient-specific clinical scenarios.

### 3.1. Primary Particle Fluence in the CONVERAY Prototype

The CONVERAY system operates using an electron beam as the primary radiation [7,8]. Typical results obtained for primary particle (electron) fluence in the CONVERAY prototype are shown in Figure 5, corresponding to the configuration of a 3 mm diameter collimation, a 1.15 mm W target, and the two dedicated deflection magnets attaining magnetic field strengths around 1 T.

Figure 6 shows the pass of the primary electron beam through the W collimator when removing the photon beam production target, as required for electron beam applications.

As appreciated in Figure 5 and Figure 6, the primary electron fluence allows us to verify the proper implementation of the magnetic field as well as its successful performance to “guide” the electron beam through the CONVERAY system.

Finally, it is worth mentioning that when rotating the CONVERAY arm at uniform angular frequency, the primary electron beam is conduced to follow the rotation and therefore to dynamically produce the convergent photon beam, as described in the Materials and Methods Section.

### 3.2. CONVERAY Phase Space and Photon Beam Production

Once the primary electron beam enters the CONVERAY prototype, it is conducted by the deflection magnets to impact perpendicularly the photon production target where the bremsstrahlung and characteristic photons traveling in the forward direction within the collimator acceptance (internal canal) can finally emerge the CONVERAY system to attain the impact surface, commonly placed close to the isocenter.

Careful characterization of the produced radiation due to the impact of the primary electron beam onto the target stands as a key issue to assess accurate dosimetry. Figure 7, Figure 8, Figure 9 and Figure 10 depict the most relevant phase state variables of the ionizing radiation emerging the CONVERAY prototype.

Particularly relevant is the energy spectrum of the emerging photons that reach the incident surface, which is reported in Figure 11.

### 3.3. CONVERAY Dosimetry Performance

The preliminary performance of the CONVERAY prototype as a feasible and valuable alternative to achieve high conformal within the target (commonly referred as PTV, or primary tumor target) dose distributions for complex patient-specific scenarios including intracranial and thoracic irradiations.

#### 3.3.1. Preliminary CONVERAY Dosimetry Performance for Intracranial Irradiations

Figure 12, Figure 13 and Figure 14 show the preliminary performance of the CONVERAY system for irradiation of a complex intracranial target.

As appreciated for the different visualization planes, Figure 11, Figure 12, Figure 13 and Figure 14 depict the expected high dose concentration around/close to the corresponding targets despite the specific collimation system.

#### 3.3.2. Preliminary CONVERAY Dosimetry Performance for Thoracic Irradiations

Figure 15 shows the preliminary performance of the CONVERAY system for thorax-level irradiation emulation.

Although promising dose conformation is obtained through irradiation with a unique static CONVERAY beam without any method/tool for treatment planning/optimization, an extra freedom degree has been incorporated with the MC simulation tool aimed at allowing different irradiation orientation, i.e., incorporating the gantry rotation, in practice.

Figure 16, Figure 17, Figure 18 and Figure 19 depict the obtained improved results for both intracranial and thoracic PTV by incorporating uniform irradiations during gantry rotations.

## 4. Discussion

The developed methodology’s feasibility and suitability for emulating the CONVERAY system using the proposed Monte Carlo simulation approach have been comprehensively assessed. The performance of the deflection magnets has been characterized, demonstrating their ability to effectively guide the incident electron beam (Figure 5 and Figure 6). Logarithmic scale fluence graphs indicate that only a minor percentage of fluence is lost, likely due to the spectral characteristics of the incident beam and minor magnetic field intensity non-uniformities. Separately, the production and collimation of the photon beam in the CONVERAY target have been successfully verified. Among various target–filter configurations studied, using a 1–2 mm thick W disc, similar to those in conventional 6 MV linacs [14], appears to be a suitable choice for producing a photon beam comparable to those from conventional linacs. Additionally, removing the target-filter set allows the electron beam to directly reach the treatment surface, enabling operation in “electron mode” (Figure 6). This operation mode might be useful for superficial treatments, whose dosimetry performance requires a dedicated study.

The phase space output file provided a detailed characterization of the emerging radiation’s relevant phase state variables. The collected information, summarized in Figure 7, Figure 8, Figure 9, Figure 10 and Figure 11, demonstrates that CONVERAY produces an intrinsic convergent photon beam focused on a central point [19,20]. Furthermore, integrating the emerging radiation over complete 360° turns revealed the expected symmetry and integral convergent capability of the CONVERAY operation mode. The phase state variable distributions enable straightforward evaluation of differences between CONVERAY configurations, such as collimation. The effects of varying rotation angle intervals or collimation diameters are illustrated in Figure 18 and Figure 19, where improved conformation can be attained according to the characteristics of the collimation system. In this regard, a small diameter combined with a long internal channel produces reduced focal spot volumes (the mean extension of the 95% isodose volume varied from 3 mm to 7 mm when the internal channel diameter varies from 3 to 10 mm, for instance).

While this proof-of-concept simulation study does not address clinical issues or dosimetry indices for treatment planning, it is noteworthy that optimizing technical parameters, such as rotation angle interval and collimator selection, can potentially lead to improved radiation delivery in terms of dose concentration and conformation within delimited target volumes. For example, carefully choosing the rotation angle interval can help minimize direct exposure to organs at risk (Figure 18 and Figure 19). Similarly, selecting an appropriate collimator can enhance dosimetric outcomes, as depicted in Figure 12 and Figure 13 for an intracranial irradiation, and Figure 19 for a thoracic irradiation.

Noticeable performance has been achieved in delivering high dose concentrations to target volumes, as demonstrated in Figure 12, Figure 13, Figure 14, Figure 15, Figure 16, Figure 17, Figure 18 and Figure 19. The unique, non-rotating CONVERAY beams exhibited characteristic in-depth ballistic capabilities along the central axis, as shown in Figure 12, Figure 13, Figure 15 and Figure 16. Furthermore, incorporating gantry rotation into the CONVERAY system, without treatment planning or optimization, yielded significant improvements in dosimetric performance, achieving preliminary acceptable target coverage and organ sparing, as illustrated in Figure 15 and Figure 19.

Future developments incorporating on-line photon beam modulation and robotic arm technology, similar to that used in CyberKnife systems [14], may enable highly accurate dose painting capabilities, potentially further enhancing the system’s performance.

Considering that the clinical significance of the CONVERAY system lies in its potential to address longstanding challenges in radiotherapy oncology as an option to integrate and/or complement existing modalities, aiming at delivering precise high-dose radiation to tumors while sparing adjacent healthy tissues to increase efficacy and reduce the risk of treatment-related toxicity, some potential clinical applications are speculatively discussed. Cancer treatment continues to rely heavily on radiation therapy, driving innovation in the development of novel modalities and techniques that enhance patient outcomes and quality of life. As treatment efficacy improves, mitigating radiation-related toxicities emerged as a critical priority [21]. Radiotherapy modalities capable of achieving a high dose gradient (fall-off) are particularly advantageous for treating deep-seated tumors of small to medium size and regular shape, as they enable precise dose delivery and minimal damage to surrounding healthy tissues [22,23]. In this regard, encouraging results have been observed in achieving high dose concentrations near or within the PTV for complex, realistic clinical cases, consistent with preliminary tests for intracranial (SRS) and thorax (lung SBRT) treatments.

The ability of a radiotherapy modality to deliver high dose concentrations with steep dose fall-off in the transverse plane is crucial for optimizing the therapeutic ratio in cancer treatment, as it enables precise tumor targeting while minimizing radiation exposure to adjacent critical structures, thereby reducing the risk of treatment-related toxicities [24,25]. Within this context, the available dosimetry capabilities are particularly advantageous in the treatment of deep-seated tumors with small-medium volumes (less than ~10 cm^3^) in the presence of adjacent critical structures, where the CONVERAY system’s dosimetry performance may offer a valuable treatment option for improving tumor control while minimizing toxicity.

Although comprehensive studies are necessary to fully assess the potential of the CONVERAY system in challenging clinical scenarios, preliminary dosimetry performance hints at promising applications. Several speculative options merit consideration: (i) Brain metastases: CONVERAY, akin to stereotactic radiosurgery (SRS), may offer a treatment option for brain metastases. High dose concentration in small-medium volumes (less than ~10 cm^3^) can facilitate lesion control [9,26,27]. (ii) Arteriovenous malformations (AVMs): CONVERAY, similar to SRS, could treat AVMs by delivering high doses to small volumes, promoting obliteration of the AVM [28]. (iii) Lung tumors: CONVERAY, analogous to stereotactic body radiation therapy (SBRT), may be suitable for small to medium-sized lung tumors, leveraging high dose concentration in small volumes for tumor control [29]. (iv) Liver metastases: CONVERAY, similar to SBRT, could treat liver metastases by delivering high doses to small volumes, facilitating lesion control [30]. (v) Spinal tumors: CONVERAY, akin to SBRT, may be applied to spinal tumors, including metastases and primary tumors, utilizing high dose concentration in small volumes for tumor control [31].

The development and implementation of advanced treatment planning algorithms and systems are crucial for optimizing radiotherapy outcomes in complex cases, as they enable the creation of highly conformal dose distributions, reduction in toxicity, and improvement in tumor control rates [26,27]. In this sense, developing a dedicated approach and associated algorithms aimed at treatment planning with the CONVERAY system appears as a key issue to improve the overall dosimetry performance to promote exhaustive quantitative comparisons against currently available commercial techniques, such as IMRT and VMAT. Thus, such a scope concentrates most of the efforts currently invested in the CONVERAY project.

## 5. Conclusions

A novel concept for convergent radiotherapy, the CONVERAY system, has been proposed and characterized through Monte Carlo simulations as a proof-of-concept. This innovative approach utilizes a focusing mechanism to produce intrinsically convergent X-ray beams, capable of high dose concentration around the convergence spot. The simulation framework has been developed and validated, demonstrating the satisfactory performance of each component and the successful incorporation of patient anatomy and treatment planning structures.

Preliminary evaluations of the CONVERAY system’s dosimetry performance demonstrated promising results in terms of target dose coverage and organ sparing. These findings suggest that the CONVERAY system has potential for achieving conformal dose concentrations. However, further development of a treatment planning algorithm that is capable of accounting for the system’s unique properties is necessary to fully explore its capabilities. Ongoing efforts are focused on refining the CONVERAY system, with the ultimate goal of integrating it with existing clinical linear accelerators.

## 6. Patents

Contents of the U.S. patent entitled “Convergent photon and electron beam generator device” ID: US20140112451A1s addressed to R. Figueroa and M. Valente have been used in this manuscript [8].

## Figures and Tables

**Figure 1 cancers-17-01189-f001:**
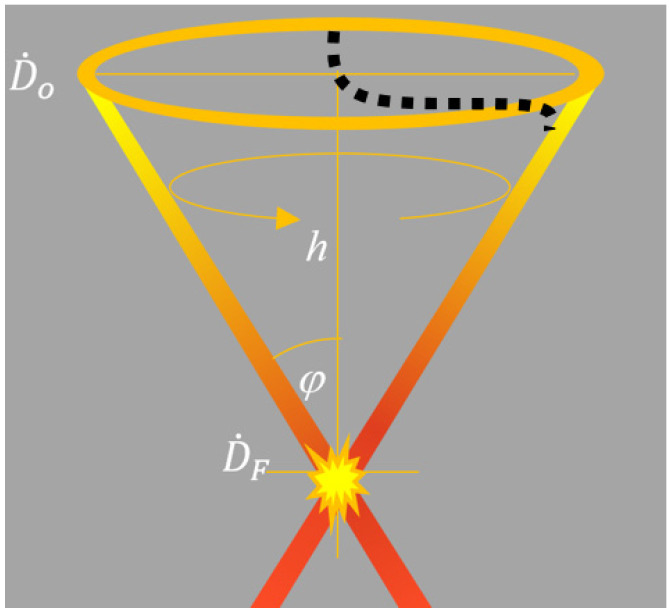
Basic sketch of convergent photon fluence in the CONVERAY system adapted from [7]. The beam propagation direction (optical axis), denoted as +*z*, corresponds to the downward vertical direction. Black dash line depicts the electron bunch trajectory.

**Figure 2 cancers-17-01189-f002:**
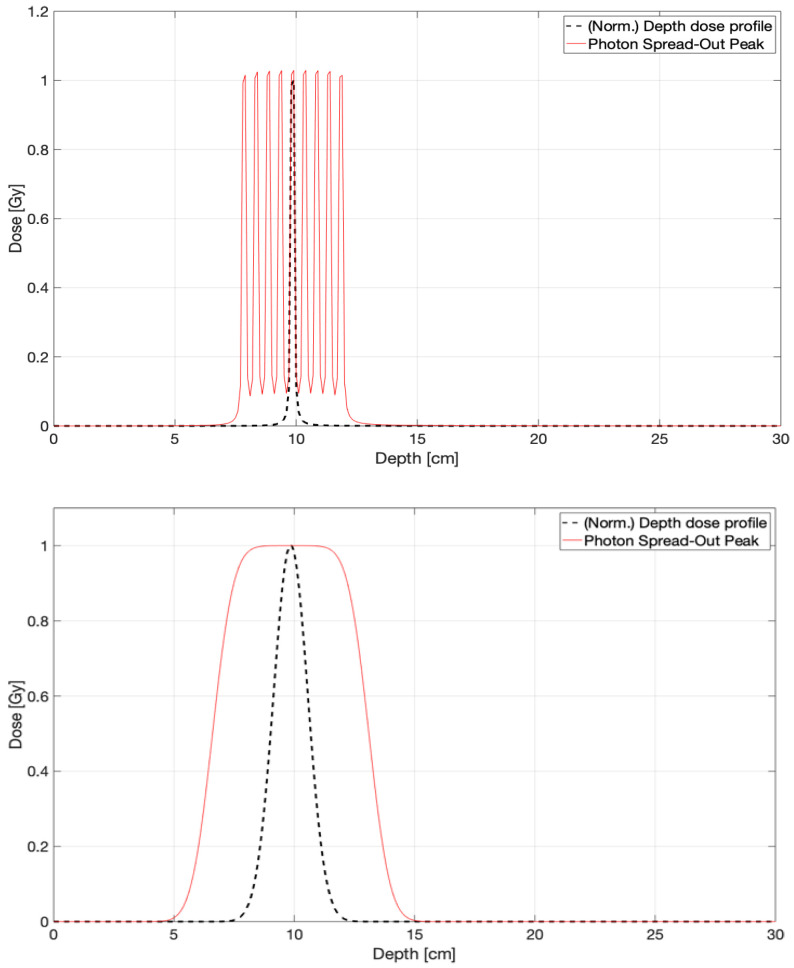
Sketch of normalized mono-energetic (**top**) and poly-energetic (**bottom**) photon spread-out peak analytically calculated by Expression (1) using the parameter set: *L* = 30 cm, *h* = 10 cm, *D*_0_ = 1 Gy, and μ = 0.0632 cm^2^/g, as it corresponds to 1.25 MeV in liquid water. The beam propagation direction (optical axis), denoted as +*z*, corresponds to the horizontal (increasing depth axis) direction.

**Figure 3 cancers-17-01189-f003:**
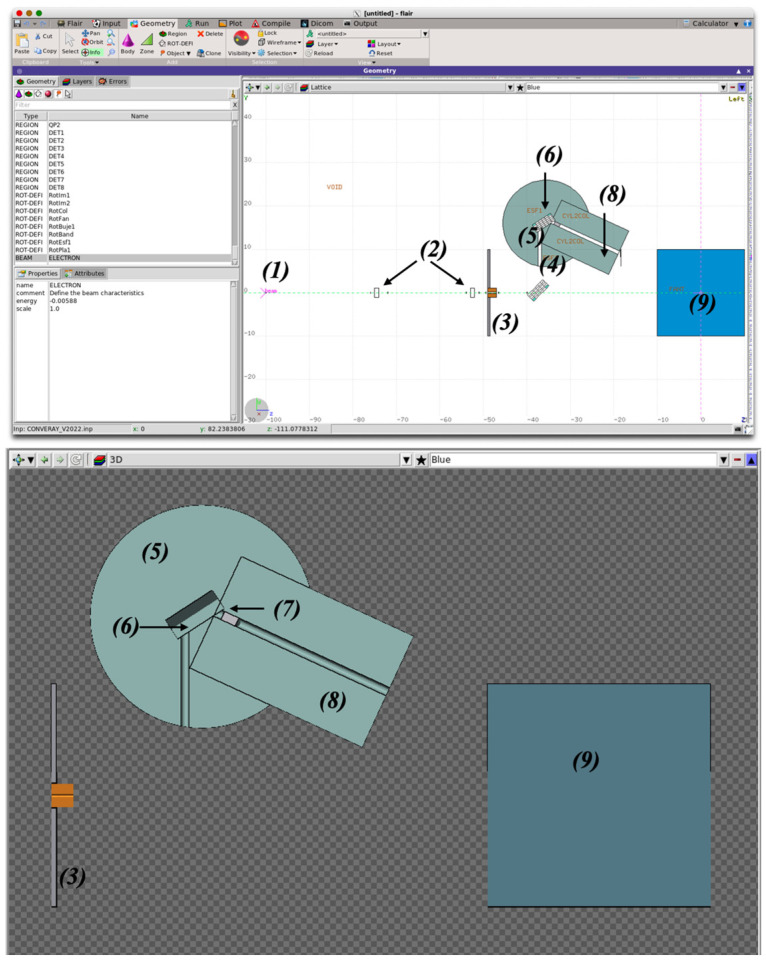
Simulation setup for the CONVERAY system, sketching the main components along with the incident electron beam and virtual detectors placed along the beam path (top) and 3D view of the system body materials irradiating a cubic water-equivalent phantom. The beam propagation direction (optical axis), denoted as +*z*, corresponds to the horizontal (from the left to the right) direction.

**Figure 4 cancers-17-01189-f004:**
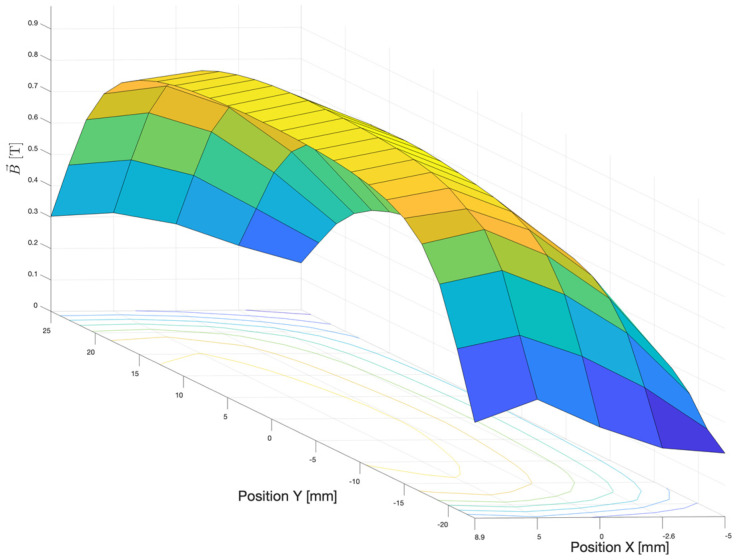
Magnetic field strength in the central plane (equidistant from the neodymium magnets) for the first (the first one chronologically attained by the incident electron beam) deflection magnet. Uncertainties in the experimental data are less than 2%. The beam propagation direction (optical axis), denoted as +*z*, enters the magnetic field according to the required deflection (tilt) angle. Surface colors indicate the magnetic field strength (from 0.1 T in blue to 1 T in yellow).

**Figure 5 cancers-17-01189-f005:**
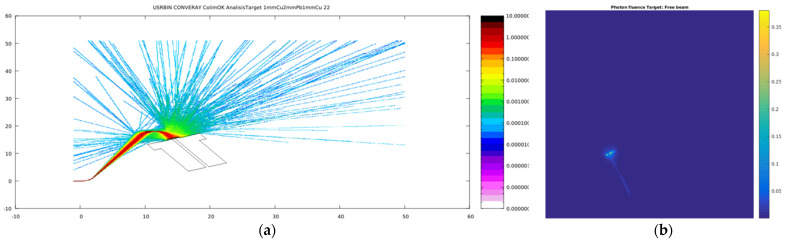
Log scale primary electron (**a**) and linear scale photon (**b**) fluence as visualized by the *FLAIR* interface and own MatLab^®^-supported processing tool, respectively. For visualization purposes, the 3 mm diameter W collimator is included as a reference piece of the CONVERAY device operating in the static way. Color bars in a logarithmic scale are illustrative, and absolute values have no specific relevance. The beam propagation direction (optical axis), denoted as +*z*, corresponds to the horizontal (from the left to the right) direction. Color-bars indicate the corresponding particle fluence. Notice that incomplete numbers in the fluence color-bar are all zero values.

**Figure 6 cancers-17-01189-f006:**
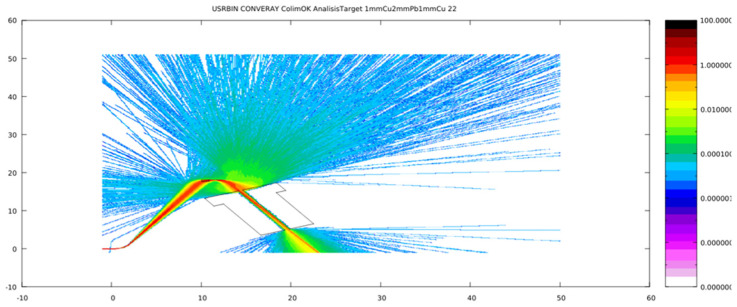
Log scale primary electron fluence as visualized by the *FLAIR* interface. For visualization purposes, the 3 mm diameter W collimator is included as a reference piece of the CONVERAY device operating in the static way. Color bars in the logarithmic scale are illustrative, and absolute values have no specific relevance. The beam propagation direction (optical axis), denoted as +*z*, corresponds to the horizontal (from the left to the right) direction. Notice that incomplete numbers in the fluence color-bar are all zero values.

**Figure 7 cancers-17-01189-f007:**
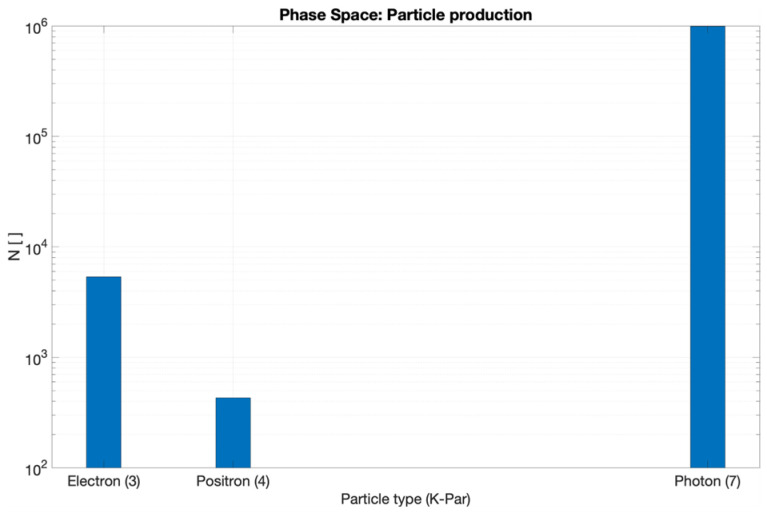
Production of different particle types at the 1.15 mm W target due to the impact of the primary electron beam conducted by the deflection magnets of the CONVERAY device.

**Figure 8 cancers-17-01189-f008:**
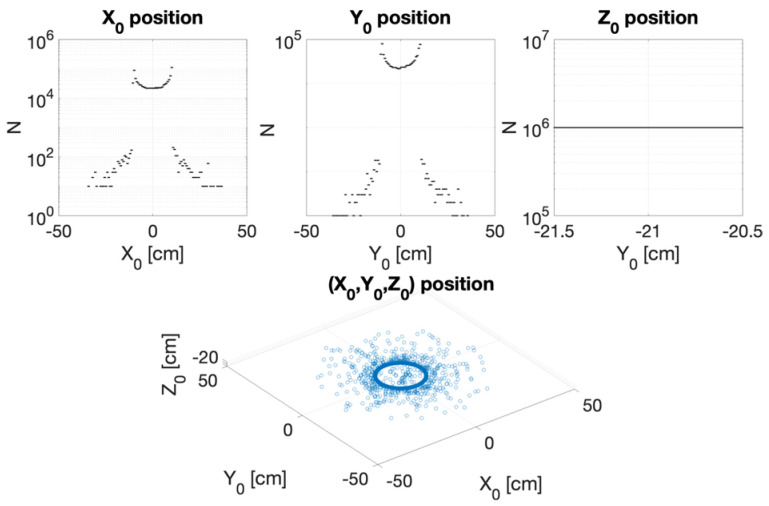
Cartesian coordinates (**top**) and 3D position (**bottom**) of ionizing radiation produced by the CONVERAY device as measured on the *xy*-plane at z = −21 cm, thus representing the radiation emerging from the dynamic CONVERAY device as integrated for a complete 360° turn.

**Figure 9 cancers-17-01189-f009:**
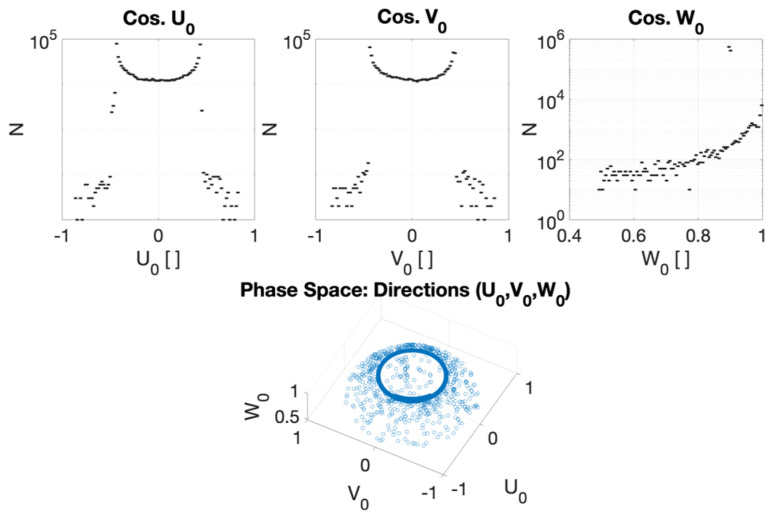
Direction cosines (**top**) and 3D (**bottom**) with respect to the positive sense of the Cartesian coordinates of the ionizing radiation emerging the CONVERAY device dynamically operating in the “photon mode” for a complete 360° turn.

**Figure 10 cancers-17-01189-f010:**
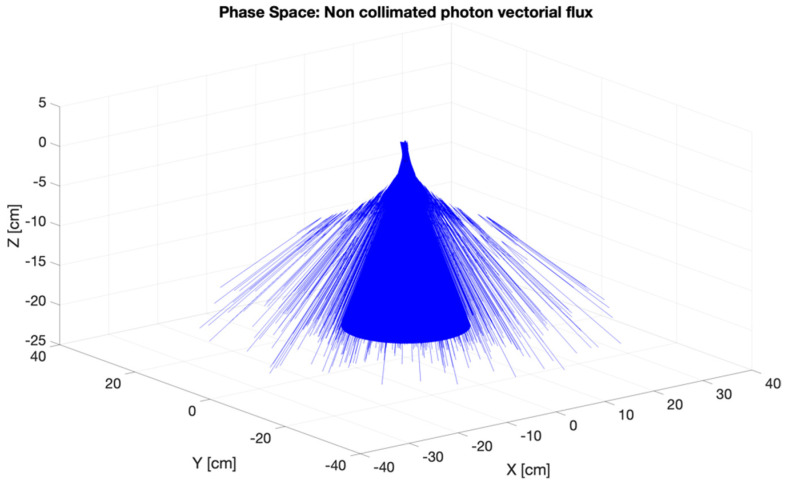
3D visualization of the vectorial flux of the ionizing radiation emerging the CONVERAY device dynamically operating in the “photon mode” for a complete 360° turn. Flow sense is in the *z*-axis increasing sense.

**Figure 11 cancers-17-01189-f011:**
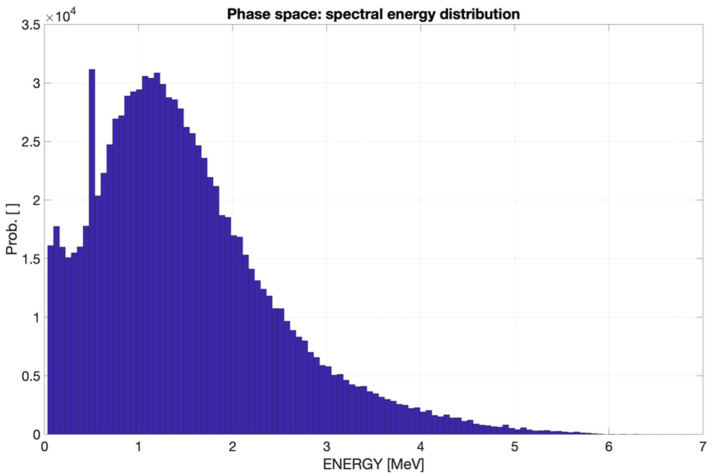
Energy spectrum of the photon beam emerging from the CONVERAY device dynamically operating in the “photon mode” for a complete 360° turn.

**Figure 12 cancers-17-01189-f012:**
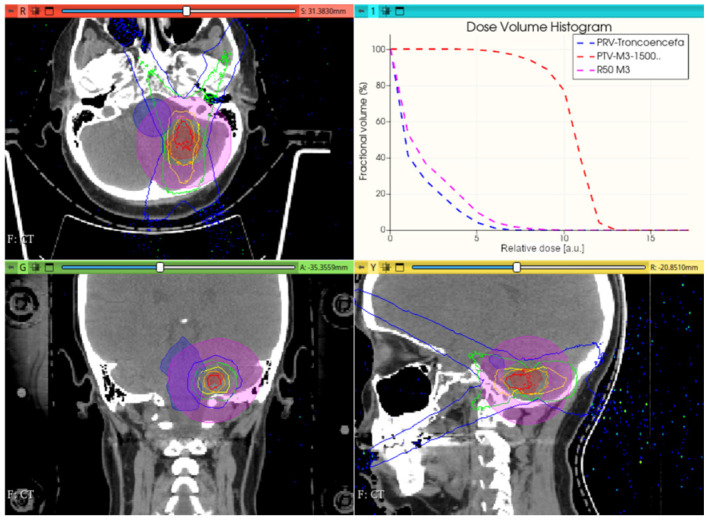
A unique (no gantry rotation) 10 mm diameter collimator CONVERAY beam attaining an intracranial PTV. Absorbed dose distribution within the three main planes along with the resulting DVH for the PTV and the more relevant organs at risk are reported. Isodose curves: 90% (red), 75% (orange), 50% (yellow), 25% (green), and 10% (blue).

**Figure 13 cancers-17-01189-f013:**
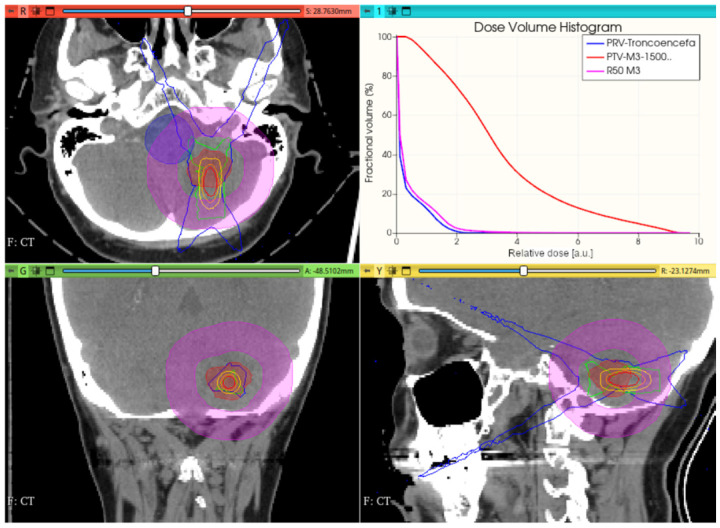
A unique (no gantry rotation) 3 mm diameter collimator CONVERAY beam attaining an intracranial PTV. Absorbed dose distribution within the three main planes along with the resulting DVH for the PTV and the more relevant organs at risk are reported. Isodose curves: 90% (red), 75% (orange), 50% (yellow), 25% (green), and 10% (blue).

**Figure 14 cancers-17-01189-f014:**
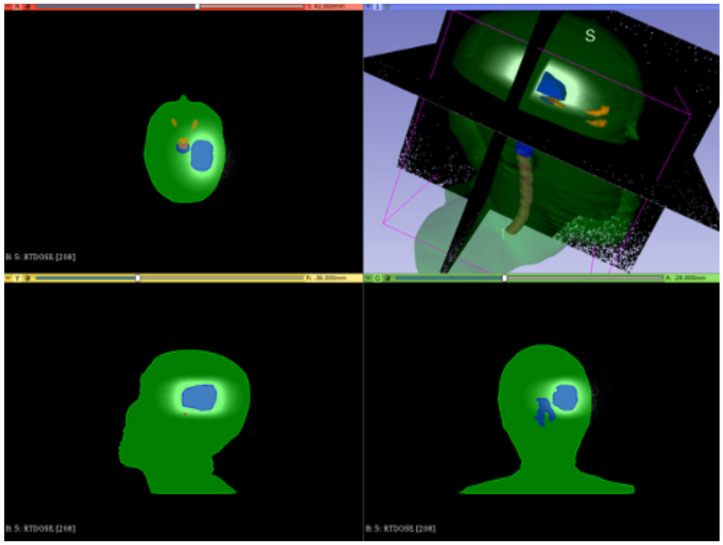
Example of dose distribution superimposed to the corresponding anatomy, as achieved by the 3D-Slicer software reporting the performance of a unique (no gantry rotation) 30 mm diameter collimator CONVERAY beam attaining an intracranial PTV highlighted in light blue.

**Figure 15 cancers-17-01189-f015:**
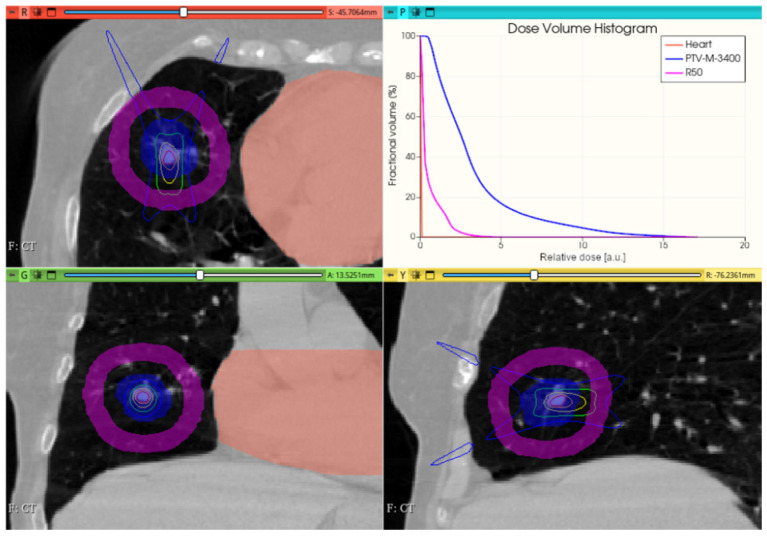
A unique (no gantry rotation) 3 mm diameter collimator CONVERAY beam attaining a thoracic PTV. Absorbed dose distribution within the three main planes along with the resulting DVH for the PTV and the more relevant organs at risk are reported. Isodose curves: 90% (red), 75% (orange), 50% (yellow), 25% (green), and 10% (blue).

**Figure 16 cancers-17-01189-f016:**
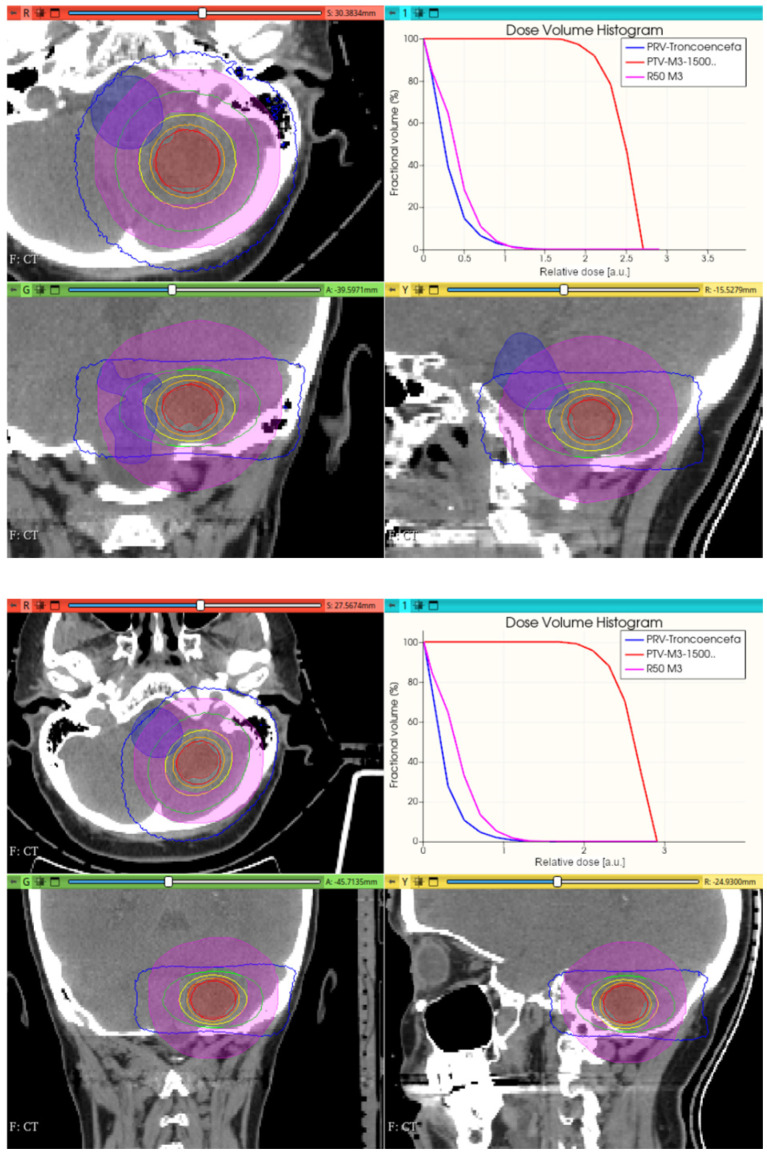
Uniform 360° (**top**) and 270° (**bottom**) gantry rotation 10 mm diameter collimator CONVERAY irradiation of an intracranial PTV. Absorbed dose distribution within the three main planes along with the resulting DVH for the PTV and the brainstem (denoted as tronco-encefa) are reported. Isodose curves: 90% (red), 75% (orange), 50% (yellow), 25% (green), and 10% (blue). Isodose curves: 90% (red), 75% (orange), 50% (yellow), 25% (green), and 10% (blue).

**Figure 17 cancers-17-01189-f017:**
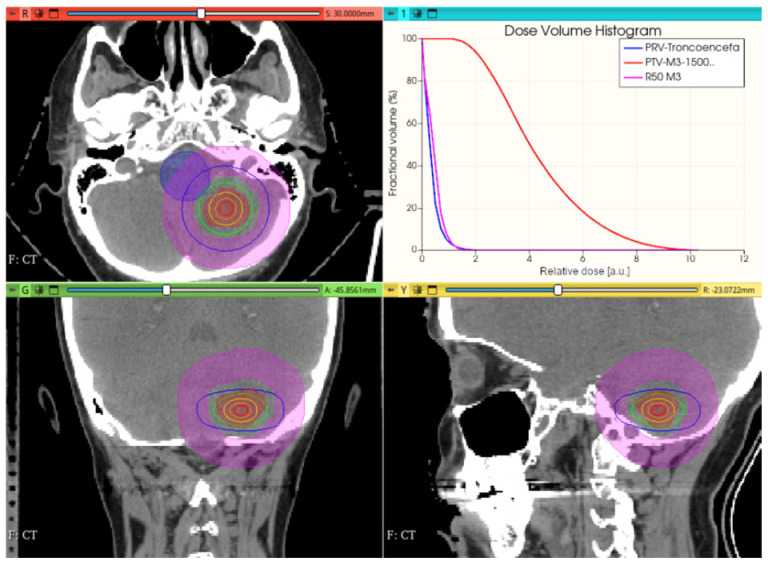
Uniform 360° gantry rotation 3 mm diameter collimator CONVERAY irradiation of an intracranial PTV. Absorbed dose distribution within the three main planes along with the resulting DVH for the PTV and the brainstem are reported. Isodose curves: 90% (red), 75% (orange), 50% (yellow), 25% (green), and 10% (blue).

**Figure 18 cancers-17-01189-f018:**
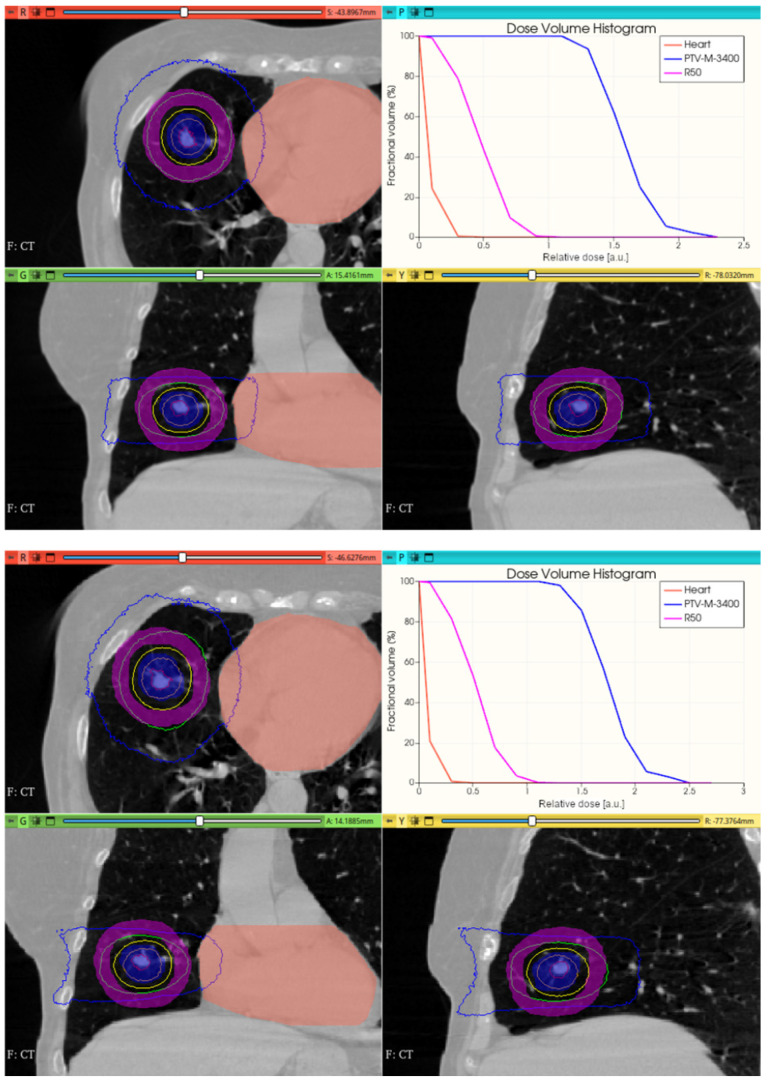
Uniform 0–360° (**top**) and 30–180° (**bottom**) gantry rotation 15 mm diameter collimator CONVERAY irradiation of a thoracic PTV. Absorbed dose distribution within the three main planes along with the resulting DVH for the PTV and the heart (denoted as *corazon*) are reported. Isodose curves: 90% (red), 75% (orange), 50% (yellow), 25% (green), and 10% (blue).

**Figure 19 cancers-17-01189-f019:**
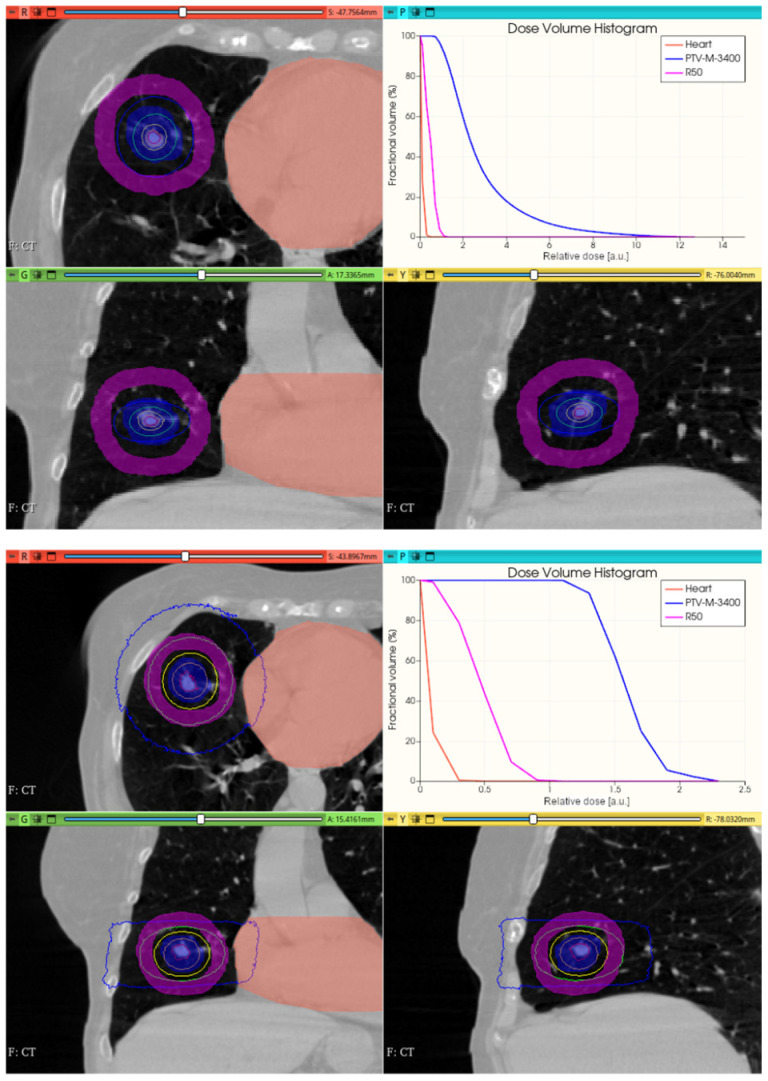
Uniform 360° gantry rotation 3 mm (**top**), and 10 mm (**bottom**) diameter collimator CONVERAY irradiation of a thoracic PTV. Absorbed dose distribution within the three main planes along with the resulting DVH for the PTV and the heart (denoted as *corazon*) along with the *R*_50_ curve are reported. Isodose curves: 90% (red), 75% (orange), 50% (yellow), 25% (green), and 10% (blue).

**Table 1 cancers-17-01189-t001:** Estimation of the uncertainties’ evolution during the convergent photon beam production as organized by the stages described in Figure 3. Values for stages (2) to (9) report the absolute value of percentage variations with respect to the initial reference case: filiform (6.0 ± 0.1) MeV electron beam. When energy values are modified, beam propagations are kept fixed (*α_beam_* = 0, *FWHM_α_* = 0); while *<E_el_>* = 6.0 MeV and *<FWHM (E_el_)>* = 0.1 MeV apply when changing the beam propagation parameters. Corresponding relative uncertainties do not exceed 5% of values reported in the table.

Variable/Stage	(1)	(2)	(3)	(4)	(5)	(6)	(7)	(8)	Final Stage
*<E_el_>*/*FWHM (E_el_)* [MeV]	6.0/0.1	-	-	-	-	-	-	-	0/0
	5.9/0.1	1.7/0	1.7/0	1.7/0	1.7/0	1.7/0	2.5/2.9	2.5/2.9	3.2/3.4
	6.1/0.1	1.6/0	1.6/0	1.6/0	1.6/0	1.6/0	2.4/2.8	2.4/2.8	3.1/3.3
	6.0/0.3	0/200	0/200	0/200	0/200	0/200	0/200	4.1/272	15.7/488
*α_βεαμ_*/*ΦΩHΜ_α_* [δεγ]	0.0/0.0	-	-	-	-	-	-	-	0/0
	0.0/0.1	0/0.9	0/1.4	0/3.6	0/0.9	6.3/14.1	10.2/38.1	11.1/49.2	16.3/64.6
	0.0/1.0	0/95.9	0/148	0/206	341/565	1735/7912	-	-	-
	1.0/0.0	2.4/1.2	2.6/1.6	3.3/59.7	11.9/241	73/618	269/6431	-	-

**Table 2 cancers-17-01189-t002:** Effects are due to simulation parameters (*E_Abs_* represents the 2-input vector for the absorption/cutoff energy for electrons and photons), while *N_Tot_* states for the total number of simulated primary histories (*showers*). Estimation of the uncertainties’ evolution during the convergent photon beam production as organized by stages described in Figure 3. Values for stages (2) to (9) report the absolute value of percentage variations with respect to the initial reference case: filiform (6.0 ± 0.1) MeV electron beam. When energy values are modified, beam propagations are kept fixed (*α_beam_* = 0, *FWHM_α_* = 0), while *<E_el_>* = 6.0 MeV and *<FWHM (E_el_)>* = 0.1 MeV apply when changing the beam propagation parameters. Corresponding relative uncertainties do not exceed 5% of values reported in the table.

Simulation Setup	*E_abs_* [eV]	*N_Tot_*	*f_t_*	∆D [%]
(6.0 ± 0.1) MeV *α_beam_* = *FWHM_α_ = 0*	(1 × 10^4^, 1 × 10^4^)	2 × 10^9^	-	-
(6.0 ± 0.1) MeV *α_beam_* = *FWHM_α_ = 0*	(1 × 10^4^, 1 × 10^4^)	2 × 10^8^	0.12	3.1%
(6.0 ± 0.1) MeV*α_beam_* = *FWHM_α_ = 0*	(1 × 10^4^, 1 × 10^4^)	2 × 10^7^	0.0118	8.6%
(6.0 ± 0.1) MeV *α_beam_* = *FWHM_α_ = 0*	(1 × 10^4^, 1 × 10^4^)	2 × 10^6^	0.001059	27.1%
(6.0 ± 0.1) MeV *α_beam_* = *FWHM_α_ = 0*	(1 × 10^4^, 1 × 10^4^)	2 × 10^5^	0.00011471	68.4%
(6.0 ± 0.1) MeV *α_beam_* = *FWHM_α_ = 0*	(1 × 10^5^, 1 × 10^4^)	2 × 10^9^	0.00001673	2.8%
(6.0 ± 0.1) MeV *α_beam_* = *FWHM_α_ = 0*	(1 × 10^3^, 1 × 10^4^)	2 × 10^9^	1.0748	2.1%
(6.0 ± 0.1) MeV *α_beam_* = *FWHM_α_ = 0*	(1 × 10^4^, 1 × 10^3^)	2 × 10^9^	1.00311	1.4%

## Data Availability

Data supporting reported results are completely available upon request.

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
