# Peer review of "Development and Validation of Monte Carlo Methods for Converay: A Proof-of-Concept Study"

_cancers, 2025, doi:10.3390/cancers17071189_

Round 1

Reviewer 1 Report

Comments and Suggestions for Authors

I would like to emphasize that I find the concept of a converging photon beam in radiotherapy particularly interesting, from an academic point of view. I would like to congratulate all the contributors to the Converay project for thinking outside of the box and focusing on the development of a new modality. Innovation is the driving force in medicine and the fight against cancer. Regarding the present manuscript, my comments are given below.

1. According to the authors, the main conclusion of this study is that the Converay system can attain high dose conformity to complex clinical targets, comparable to modern radiotherapy techniques (e.g., lines 33-36 and 420-426). In my opinion, this is an over-statement. I question these claims by the authors. For example, looking at the DVHs in Figs 12-19, I do not believe that they are clinically acceptable. Given the high spatial and dosimetric requirements of SRS and SBRT, the DVHs for the targets are far from being clinically acceptable (especially those in Figs 13, 15 and 17). I acknowledge that they were produced without any treatment planning, but still from a clinical perspective the deduced dose distributions cannot be delivered to a patient. Thus, the main scope (and conclusion) of this manuscript is not served.

2. The authors repeatedly claim that the derived dose distributions are comparable to those from modern radiotherapy treatment platforms for SRS and SBRT. However, there is no relevant comparison in the manuscript to be able to determine that.

3. This is purely a simulation study, using two well-established Monte Carlo codes (FLUKA and PENELOPE). However, important technical details have been omitted by the authors. As an instance, a detailed dosimetric uncertainty analysis is missing which should include both statistical (Type A) and systematic (Type B, e.g., cross section uncertainty, material and geometric tolerances, electron transport uncertainty, etc.). Moreover, simulation parameters are not mentioned. Variance reduction techniques? Accuracy of the electron transport algorithm (e.g., Fano test results)? Optimization of the Monte Carlo simulations to strike a balance between accuracy and efficiency? To sum up, the authors greatly rely on Monte Carlo results, but the methodology to derive the calculated dose distributions is poorly written.

4. Dose-volume metrics are not given. Clinical acceptability of a dose distribution cannot solely rely on DVHs. For example, is the V12Gy dose constraint for the normal brain parenchyma met? What about lung dose-volume constraints? It is important to present achieved dose-volume indices, from a clinical perspective.

5. Moreover, I strongly disagree with the statement that this is “a feasibility and suitability study of the developed methodology… has been exhaustively evaluated” (e.g., lines 359-361). The evaluation of the methodology was far from being exhaustive. In addition, this is not a feasibility or suitability study. It can be regarded more as proof-of-concept. In my opinion, the authors should have focused only on evaluating and optimizing the Monte Carlo methods for the  Converay device, rather than jumping directly to clinical suitability.

6. In my opinion, references to particle therapy (even in the Abstract and the Introduction) are not relevant and should be removed. I understand that an analogy with the spread-out Bragg peak is  implied here, but I think that this is not appropriate in a scientific paper.

7. The historical overview of radiotherapy given in lines 57-65 is redundant and can be removed. In the Introduction section, I strongly recommend focusing more on the technical description of the Converay system and the current status of the project.

8. Eq 1 and Fig 1: All variables and parameters shown here should be defined and explained in the text.

9. The Discussion and Conclusion sections include many over-statements that cannot be derived/supported from the presented results. I suggest that the authors refrain from making too general and vague statements.

Author Response

I would like to emphasize that I find the concept of a converging photon 
beam in radiotherapy particularly interesting, from an academic point of 
view. I would like to congratulate all the contributors to the Converay 
project for thinking outside of the box and focusing on the development 
of a new modality. Innovation is the driving force in medicine and the 
fight against cancer. Regarding the present manuscript, my comments 
are given below.

ANSWER/REBUTTAL: The authors highly appreciate the eQorts during the evaluation 
along with general and specific comments from the Reviewer #2 highlighting the 
interests on converging photon beam radiotherapy, as well as corresponding valuable 
suggestions that have been carefully considered and implemented to improve the 
original version of the manuscript.

3. According to the authors, the main conclusion of this study is that the 
Converay system can attain high dose conformity to complex clinical 
targets, comparable to modern radiotherapy techniques (e.g., lines 33-
36 and 420-426). In my opinion, this is an over-statement. I question 
these claims by the authors. For example, looking at the DVHs in Figs 
12-19, I do not believe that they are clinically acceptable. Given the high 
spatial and dosimetric requirements of SRS and SBRT, the DVHs for the 
targets are far from being clinically acceptable (especially those in Figs 
13, 15 and 17). I acknowledge that they were produced without any 
treatment planning, but still from a clinical perspective the deduced 
dose distributions cannot be delivered to a patient. Thus, the main 
scope (and conclusion) of this manuscript is not served.

ANSWER/REBUTTAL: The authors thank the Reviewer #2 for pointing out the issue. In 
fact, claiming about the comparative performance of the CONVERAY system to 
existing modern radiotherapy appeared as -at least- too enthusiastic and, strictly, not 
properly supported by the reported results. At this point it should be emphasized that 
this query (item 3) has been addressed within the context involving the Editor’s 
general comments as well as other suggestions from Reviewers relevant to the topic
(as item 8). Thereby, such statements not fully supported by results obtained in the 
current preliminary proof-of-concept study, have been avoided. Moreover, any 
reference to clinical considerations has been removed to focus the manuscript on the 
main useful and valuable outcomes from the modelling approach. This query from the 
Reviewer #2 has been particularly welcome to provide proper context to the actual 
need for a treatment planning/optimization process before going deeper in dosimetry 
performance at clinical level. Thus, addressing the development of such a 
methodology/tool constitutes now one of the main scopes for the CONVERAY project. 

4. The authors repeatedly claim that the derived dose distributions are 
comparable to those from modern radiotherapy treatment platforms for 
SRS and SBRT. However, there is no relevant comparison in the 
manuscript to be able to determine that.

ANSWER/REBUTTAL: The authors thank for remarking this issue found many times in 
the original manuscript. According to the previous item along with the general context 
re-organizing the manuscript as a proof-of-concept, claims regarding comparative 
performance have been removed. Such statements shall be further investigated in 
detail during future CONVERAY project stages dedicated to exhaustive comparisons 
against existing radiotherapy techniques.

5. This is purely a simulation study, using two well-established Monte 
Carlo codes (FLUKA and PENELOPE). However, important technical 
details have been omitted by the authors. As an instance, a detailed 
dosimetric uncertainty analysis is missing which should include both 
statistical (Type A) and systematic (Type B, e.g., cross section 
uncertainty, material and geometric tolerances, electron transport 
uncertainty, etc.). Moreover, simulation parameters are not mentioned. 
Variance reduction techniques? Accuracy of the electron transport 
algorithm (e.g., Fano test results)? Optimization of the Monte Carlo 
simulations to strike a balance between accuracy and eQiciency? To 
sum up, the authors greatly rely on Monte Carlo results, but the 
methodology to derive the calculated dose distributions is poorly 
written.

ANSWER/REBUTTAL: The authors are particularly grateful for pointing out this issue. 
In fact, the authors agree that being Monte Carlo simulation the main methodological 
support of this study, relevant information was missed/omitted in the original version. 
In this regard, the authors have followed this query from Reviewer #2, thus including a 
complete description about the involved uncertainties along with corresponding trend 
during the convergent photon beam production. Please, refer to the subsection 2.3 
Uncertainties of the CONVERAY Monte Carlo modelling devoted to present detail 
descriptions of the implemented approach to assess the accuracy level of the 
modelling process according along with an overall uncertainties’ assessment. 
6. Dose-volume metrics are not given. Clinical acceptability of a dose 

distribution cannot solely rely on DVHs. For example, is the V12Gy dose 
constraint for the normal brain parenchyma met? What about lung 
dose-volume constraints? It is important to present achieved dosevolume indices, from a clinical perspective.

ANSWER/REBUTTAL: The authors agree with the Reviewer #1 comment, DV metrics 
were not provided in the original version of the manuscript. As it might be expected, 
lacking a tool/system to plan/optimize the treatment strongly limits the quantitative 
dosimetry performance. This issue should be addressed within the context of items 
1), 8), a d 13) of this list, considering the current manuscript re-organization focused 
on a proof-of-concept all clinical relevant indices, such as dose-volume metrics, are 
not more appropriate to be accounted for and reported herein. 

7. Moreover, I strongly disagree with the statement that this is “a feasibility 
and suitability study of the developed methodology… has been 
exhaustively evaluated” (e.g., lines 359-361). The evaluation of the 
methodology was far from being exhaustive. In addition, this is not a 
feasibility or suitability study. It can be regarded more as proof-ofconcept. In my opinion, the authors should have focused only on evaluating and optimizing the Monte Carlo methods for the Converay 
device, rather than jumping directly to clinical suitability. J.G.

ANSWER/REBUTTAL: The authors completely agree with this query. According to the 
revised version re-organization, focus is directed only on technical feasibility features 
along with preliminary performance for diose concentration. Thereby, any statement 
like “…a feasibility and suitability study of the developed methodology… has been 
exhaustively evaluated …” are removed, while descriptions about the methodology are 
significantly improved aimed at providing required support for the current “proof-ofconcept” organization. The authors are especially grateful to the Reviewer #1 for 
suggestion such an approach for the manuscript structure.

8. In my opinion, references to particle therapy (even in the Abstract and 
the Introduction) are not relevant and should be removed. I understand 
that an analogy with the spread-out Bragg peak is implied here, but I 
think that this is not appropriate in a scientific paper.

ANSWER/REBUTTAL: The authors are grateful for the Reviewer #1 suggestion. It is 
worth mentioning that this query (item 8) has been addressed within the framework of 
the main Editor’s requirements (item 1) along with suggestions from the Reviewer #2 
(item 13) from this list. Additionally, the Introduction section has been completely 
revised and re-organized removing all claims involving particle therapy.

9. The historical overview of radiotherapy given in lines 57-65 is redundant 
and can be removed. In the Introduction section, I strongly recommend 
focusing more on the technical description of the Converay system and 
the current status of the project.

ANSWER/REBUTTAL: The authors thank the Reviewer #1 for this suggestion that has 
been considered and implemented in the revised manuscript. Particularly, the 
paragraph about a historical radiotherapy overview has been removed accordingly. 

10.Eq 1 and Fig 1: All variables and parameters shown here should be 
defined and explained in the text.

ANSWER/REBUTTAL: The authors thank the Reviewer #1 for this query. Unfortunately, 
some quantities were not properly introduced/defined in the original manuscript 
version. The issue has been addressed and properly solved in the revised version

11. The Discussion and Conclusion sections include many over-statements 
that cannot be derived/supported from the presented results. I suggest 
that the authors refrain from making too general and vague statements.

ANSWER/REBUTTAL: The authors appreciated and welcome this query required by 
the Reviewer #1, because it provided the chance to revise, correct, and improve the 
Discission and Conclusions sections. Please, notice that some of the main 
modifications carried out to correct and improve the text in the original version of the 
Discussion section, as suggested by the Reviewer #1: (Examples of sentence removed that contained, originally, over-statements)

The feasibility and suitability of the developed methodology to emulate the 
CONVERAY system by means of the proposed Monte Carlo simulation approach have 
been exhaustively evaluated. …

All relevant phase state variables of the emerging radiation have been detailed 
characterized by means … 

Although quantitative clinical issues and dosimetry indices are out of scope for the 
present preliminary study, it should be noticed that significant improvements in 
organ at risk protection along with treatment time (beam on) reduction are 
attainable by implementing a simple task like properly selecting the rotation angle 
interval to avoid OARs direct exposure, as … 

Similarly, a suitable choice of the collimator can significantly benefit the clinical 
dosimetry outputs, as indicated …

Noticeable performance has been obtained to achieve high dose concentrations 
close to or inside the PTV for complex realistic clinical cases, as … 

On the other hand, the addition the gantry rotation degree of freedom to the 
CONVERAY® system, even without any kind of treatment planning/optimization 
procedure, has demonstrated significant improvements in terms of quantitative 
clinical dosimetry performance, attaining preliminary acceptable PTV coverage
and OARs protection, as 

Reviewer 2 Report

Comments and Suggestions for Authors

This paper summarizes the fundamental principles for describing converging beams dosimetry performance, along with the application to the CONVERAY system and a preliminary study on clinical applications. The language is adequate and the subject under investigation is relevant with the journal’s thematology and would be of interest to the readership. I suggest a minor revision prior to publication of the manuscript.

General Comment to Authors:

Since in this this study no physical measurements have been performed, I would suggest the authors make it more evident in the title and the introduction, maybe in the aim of the study too, that this is a preliminary study performed with Monte Carlo simulations.

Additionally, even if the study is preliminary and quantitative results are out of its scope, since the authors make comparisons and comments on its performance, some indices even from the calculated DVHs should be provided to further support their claims.

Minor Comments:

 Abstract (Lines 30-33): The results section is quite general and is closer to the methodology’s description. This is relevant to the general comments I have provided.

 Introduction (Lines 106-114): This section of the introduction resembles an outcome of the research and does not belong here; it seems more like a conclusion.

 Methods & Materials (Line 154): “showing the main components”: It would be helpful for the readers if each Latin number was indicated in Figure 3 to further explain the components.

 Results (Line 325): “…without any treatment planning”. Could the authors further explain what this phrase includes? Treatment planning in a system involves the beam geometry configuration, the optimization (optionally) and the dose calculation.

 Discussion (Lines 384-389): Qualitative results may not be adequate to perform comparisons on the improvements provided by this technique. Specifically, the dose distributions have just been presented and not compared with other irradiation techniques. Additionally, further explanation is required since the results from the intercomparison between the different collimators are not indicative of the benefits from each one.

Author Response

This paper summarizes the fundamental principles for describing 
converging beams dosimetry performance, along with the application to 
the CONVERAY system and a preliminary study on clinical applications. 
The language is adequate and the subject under investigation is relevant 
with the journal’s thematology and would be of interest to the 
readership. I suggest a minor revision prior to publication of the 
manuscript.
General Comment to Authors: Since in this this study no physical 
measurements have been performed, I would suggest the authors make 
it more evident in the title and the introduction, maybe in the aim of the 
study too, that this is a preliminary study performed with Monte Carlo 
simulations.
Additionally, even if the study is preliminary and quantitative results are out of its 
scope, since the authors make comparisons and comments on its performance, 
some indices even from the calculated DVHs should be provided to further support 
their claims.

ANSWER/REBUTTAL: The authors highly appreciate the eQorts and valuable tome 
invested by the Reviewer #2 to provide very useful feedback containing key features to 
improve the original manuscript. The authors addressed this query together with 
items 1), 8) a d 13) of this list and within the context -following the Reviewer #2 
suggestion- of implementing a significant manuscript re-organization to present the 
study as a MC-based proof-of-concept. Accordingly, the title for the revised version, 
aimed at reflecting the actual characteristics of the manuscript content is: 
“Development and Validation of Monte Carlo Methods for Converay: A Proof-ofConcept Study”
Minor Comments:

13.Abstract (Lines 30-33): The results section is quite general and is closer 
to the methodology’s description. This is relevant to the general 
comments I have provided.

ANSWER/REBUTTAL: The authors thank the Reviewer #2 for this valuable suggestion. 
It is worth mentioning that this query (item 13) has been addressed along with items 1 
and 8 from this list with the aim of providing integral fulfilments of requirements from 
the Editor and the Reviewers.
The text in lines 30-33 (brief description in the abstract of the main obtained results) of 
the original manuscript version has been completely modified. Thus, the brief 
description of the main results within the abstract in the revised has been changed as 
follows:
(text in the original version): The dosimetry performance on clinical conditions is 
reported for representative intracranial and chest irradiations. Spatial dose 
distributions along with dose-volume histograms have been calculated and compared with the corresponding clinical treatment plans
(text in the revised version): Monte Carlo simulations successfully tracked the phase 
state of the CONVERAY device, characterizing the influence of individual components 
on convergent photon beam production. Simulations evaluated dosimetry 
performance, confirming the device's capability to achieve high dose concentrations 
around the focal spot. Preliminary tests on realistic scenarios (intracranial and 
pulmonary irradiations) demonstrated promising spatial dose concentration within 
tumor volumes, while gantry rotation significantly improved dose conformation.

14.Introduction (Lines 106-114): This section of the introduction resembles 
an outcome of the research and does not belong here; it seems more 
like a conclusion.

ANSWER/REBUTTAL: The authors welcome the Reviewer #2 suggestion. It is worth 
pointing out that the Introduction section has been quite completely re-structured. 
Although some information - minimum necessary issues- originally described in lines 
106-114 has been maintained, authors are completely open to reduced/remove it 
upon suggestion/requirement. 

15.Methods & Materials (Line 154): “showing the main components”: It 
would be helpful for the readers if each Latin number was indicated in 
Figure 3 to further explain the components.

ANSWER/REBUTTAL: The authors receive this suggestion as a very valuable issue. 
Thus, the Figure 3 has been accordingly modified to provide a better interpretation 
linking the figure visualization with the corresponding text.

16.Results (Line 325): “…without any treatment planning”. Could the 
authors further explain what this phrase includes? Treatment planning 
in a system involves the beam geometry configuration, the optimization 
(optionally) and the dose calculation.

ANSWER/REBUTTAL: The authors would like to comment that such a statement has 
been removed from the manuscript. Nonetheless, the sentence had been included 
with the main purpose -clearly not achieved- of stating that irradiation have been 
performed pointing the CONVERAY beams onto the targets, i.e. no tool/methodology 
had assisted to plan/optimize the irradiation setup. As might be suspected, lacking 
such a tool represents serious limitations to evaluate the definitive dosimetry 
performance. Thereby, the re-organized manuscript focuses on preliminary proof-ofconcept tests limited to verify the capability to attain high dose concentration within 
complex targets, once a MC simulation tool is properly adapted to model the basic 
and operational performance of the CONVERAY system. The authors have considered 
this query to reinforce the already planned project stage devoted to design and 
implement some tool to plan/optimize CONVERAY irradiations. Corresponding 
outcomes shall be very valuable information about the CONVERAY system and, if 
applicable, shall be part of future project communications.

17.Discussion (Lines 384-389): Qualitative results may not be adequate to 
perform comparisons on the improvements provided by this technique. 
Specifically, the dose distributions have just been presented and not 
compared with other irradiation techniques. Additionally, further 
explanation is required since the results from the intercomparison 
between the diQerent collimators are not indicative of the benefits from 
each one.

ANSWER/REBUTTAL: The authors are grateful to the Reviewer #2 for pointing out this 
issue. It is worth mentioning that this query has been addressed and solved together 
with items 1), 8), and 13), of this list within the context of a main manuscript re-
organization. Then, the text (L384-L389) of the original version of the Discussion 
section [“Although quantitative clinical issues and dosimetry indices are out of scope 
for the present preliminary study, it should be noticed that significant improvements in 
organ at risk protection along with treatment time (beam on) reduction are attainable 
by implementing a simple task like properly selecting the rotation angle interval to 
avoid OARs direct exposure, as appreciated in Figures 18 and 19. Similarly, a suitable 
choice of the collimator can significantly benefit the clinical dosimetry outputs, as 
indicated in Figures 12 and 13 for SRS, and 19 for lung SBRT.”] has been removed. 
Moreover, the revised version provides some insights about the eQects due to the 
diQerent collimation systems (sizes). A brief comment is provided in the revised man 
uscript ab out the influence of the collimation: “ … The phase state variable 
distributions enable straightforward evaluation of diZerences between CONVERAY 
configurations, such as collimation. The eZects of varying rotation angle intervals or 
collimation diameters are illustrated in Figures 18 and 19, where improved 
conformation can be attained according to the characteristics of the collimation 
system. In this regard, small diameter combined with long internal channel produces 
reduced focal spot volumes (the mean extension of the 95% isodose volume varied 
from 3 mm to 7 mm when the internal channel diameter varies from 3 to 10 mm, for 
instance)” [L446-L453]. As appreciated, the 95% isodose volumes for the same virtual 
patient, but diQerent collimations have been calculated and reported

Reviewer 3 Report

Comments and Suggestions for Authors

This is a comment on the work done by numbered cancers-3351983 entitled CONVERAY: A feasibility study for an innovative convergent irradiation technique” submitted to Cancers (ISSN 2072-6694)

The authors aimed to work on a study called as CONVERAY proposes an innovative teletherapy system based on a convergent X-ray beam and evaluates its dosimetry performance through Monte Carlo simulations for representative intracranial and chest irradiations. It can produce a convergent X-ray beam that achieves highly conformal dose distributions to complex clinical targets, as demonstrated through dosimetry simulations for SRS and SBRT cases. Having been adapted to current existing linear accelerators, providing an attractive alternative to highly conform dose distributions, which enables the high dose conformation. In summary, the authors tried to improve treatment accuracy and reduce side effects by their study. Furthermore, the study presents preliminary dosimetry performance results, which may not fully capture the complexities of clinical applications. The authors indicate that quantitative clinical issues and dosimetry indices are beyond the scope of this initial study, suggesting that further research is required to validate the findings in real-world scenarios.

Although the paper does not present a novel hypothesis about a related research field and it only tested from a limited application with preliminary attempt, I do not know whether preliminary studies fit with journal’s standard, it experimentally tests findings and fresh experiments over the related literature. It can be beneficial to the related literature and contribute to enriching the literature on such subjects. It could be seen that different experiments were carried out in the study, and it is a well-presented paper. Even though the copy and screen paste photos are used a lot in the images in the work, it is open to see nice experimental effort over the study.

I liked the work, but the authors are expected to provide a comprehensive response to the following questions.

This manuscript could be a good candidate for the journal, provided that the necessary corrections are made in accordance with the editor's ultimate decision.

-The accuracy of the dosimetric calculations presented herein is contingent upon the fidelity of the Monte Carlo (MC) simulations employed. Although MC methods are a powerful tool for simulating radiation transport and interaction, they are inherently subject to uncertainties arising from the underlying physical models, approximations in the implementation of these models, and the specific assumptions adopted within the simulation framework. It is essential to recognize that the performance characteristics of critical components, including the deflection magnets and other relevant elements within the beamline, were determined through simulation. Although the effort has been made to guarantee the precision of these simulations, discrepancies may arise between the simulated and actual operational parameters of these components, potentially impacting the final dosimetric accuracy. Further experimental validation of the simulated component performance may be necessary to quantify the impact of the potential deviations. This could enhance confidence in the reported dosimetric results and provide a more robust assessment of the suggested MC’s performance in a realistic operational situation.

-Dosimetric inaccuracies may arise due to fluence variations coming from the incident beam's spectral characteristics and spatial non-uniformities in the magnetic field intensity. Although these fluence deviations are anticipated to be minimal, at the order of a few percent, they have the potential to compromise both the accuracy of treatment planning calculations and the overall clinical efficacy of the delivered dose. Further investigation is required to quantify the magnitude of these effects and to develop, if necessary, mitigation strategies to ensure consistent and predictable dose delivery. This may require the implementation of advanced techniques for the characterization of the beam spectrum and magnetic field profile, in conjunction with the refinement of treatment planning algorithms to account for these non-ideal conditions. The objective is to achieve the highest level of dosimetric accuracy and treatment efficacy, thus ensuring optimal patient outcomes.

- The research didn't fully integrate treatment planning and optimization algorithms, which are crucial for ensuring clinical effectiveness. The difficulties of individualized dosimetry and delivery optimum dose strategies are crucial for minimizing off-target effects and maximizing targeted dose deposition within the clinical treatment paradigm. This represents a significant limitation, impeding a comprehensive assessment of the effects under realistic clinical conditions. Further studies are required to incorporate these critical elements so as to evaluate the clinical translatability of the observed effects.

-In my view, the analysis and contextualization of the present findings within the existing body of literature should be some strengthen. The discussion section currently lacks sufficient depth in comparing the results with relevant peer research, limiting the study's contribution to the field. Because this study is a research article rather than a technical report, it would be advantageous to expand the discussion section.

As a scientific reader, I should find response to some of my questions as a reader of this article.

Given the manuscript's intended format as a research article, a more comprehensive discussion looks crucial. This necessitates a more thorough comparison with established literature and interpretation of the findings, exploring potential implications, limitations, and avenues for future research. Expanding this section could elevate the manuscript's scientific rigor and a more impactful contribution to the broader scholarly. Specifically, the discussion should address: How do the findings agreements or contradict previous works? (e.g., methodological differences, statical variations)? How do the results inform with existing MC frameworks? Do authors suggest new MC directions? How might these limitations affect the interpretation of the findings? Table presentation could be beneficial for interpretation of some results? So, the results lack statistical analysis and some of the results should be tabulated.

Author Response

This is a comment on the work done by numbered cancers-3351983 
entitled CONVERAY: A feasibility study for an innovative convergent 
irradiation technique” submitted to Cancers (ISSN 2072-6694).

The authors aimed to work on a study called as CONVERAY proposes an 
innovative teletherapy system based on a convergent X-ray beam and 
evaluates its dosimetry performance through Monte Carlo simulations 
for representative intracranial and chest irradiations. It can produce a 
convergent X-ray beam that achieves highly conformal dose 
distributions to complex clinical targets, as demonstrated through 
dosimetry simulations for SRS and SBRT cases. Having been adapted to 
current existing linear accelerators, providing an attractive alternative to 
highly conform dose distributions, which enables the high dose 
conformation. In summary, the authors tried to improve treatment 
accuracy and reduce side eQects by their study. Furthermore, the study 
presents preliminary dosimetry performance results, which may not 
fully capture the complexities of clinical applications. The authors 
indicate that quantitative clinical issues and dosimetry indices are 
beyond the scope of this initial study, suggesting that further research is 
required to validate the findings in real-world scenarios.
Although the paper does not present a novel hypothesis about a related 
research field and it only tested from a limited application with 
preliminary attempt, I do not know whether preliminary studies fit with 
journal’s standard, it experimentally tests findings and fresh 
experiments over the related literature. It can be beneficial to the 
related literature and contribute to enriching the literature on such 
subjects. It could be seen that diQerent experiments were carried out in 
the study, and it is a well-presented paper. Even though the copy and 
screen paste photos are used a lot in the images in the work, it is open 
to see nice experimental eQort over the study.
I liked the work, but the authors are expected to provide a 
comprehensive response to the following questions.
This manuscript could be a good candidate for the journal, provided that 
the necessary corrections are made in accordance with the editor's 
ultimate decision.

ANSWER/REBUTTAL: The authors are grateful to the Reviewer #3 for investing so 
dedicated eQorts to review the manuscript. All queries and concerns have been taken 
into account and implemented to correct and improved the manuscript. The authors 
are highly motivated by the Reviewer #3 recognition about the experimental eQorts 
underlying this simulation study. First, the authors would like to comment that queries 
& feedback in the general comments from the Reviewer #3 should be contextualized 
within the overall revision/correction process. In this regard, the revised version is 
focused on reporting a proof-of-concept study, thus avoiding statements linked with 
pursuing a definitive characterization of the dosimetry performance for clinical 
applications. Therefore, claims about quantitative clinical issues and dosimetry 
indices are eQectively, beyond the scope of the revised manuscript aimed at a MCbased proof-of-concept. Further developments might drive the project progresses to 
the next exciting stages. 

19. The accuracy of the dosimetric calculations presented herein is 
contingent upon the fidelity of the Monte Carlo (MC) simulations 
employed. Although MC methods are a powerful tool for simulating 
radiation transport and interaction, they are inherently subject to 
uncertainties arising from the underlying physical models, 
approximations in the implementation of these models, and the specific 
assumptions adopted within the simulation framework. It is essential to 
recognize that the performance characteristics of critical components, 
including the deflection magnets and other relevant elements within the 
beamline, were determined through simulation. Although the eQort has 
been made to guarantee the precision of these simulations, 
discrepancies may arise between the simulated and actual operational 
parameters of these components, potentially impacting the final 
dosimetric accuracy. Further experimental validation of the simulated 
component performance may be necessary to quantify the impact of 
the potential deviations. This could enhance confidence in the reported 
dosimetric results and provide a more robust assessment of the 
suggested MC’s performance in a realistic operational situation.

ANSWER/REBUTTAL: The authors are very grateful to the Reviewer #3 for remarking 
this issue, which has been addressed along with previous items of this list regarding 
the same core feature. Please, refer to the subsection 2.3 Uncertainties of the 
CONVERAY Monte Carlo modelling that has been added to fulfil the aforementioned 
weakness regarding the eQects on the overall uncertainties due to critical setup and 
operational parameters. 

20.Dosimetric inaccuracies may arise due to fluence variations coming 
from the incident beam's spectral characteristics and spatial nonuniformities in the magnetic field intensity. Although these fluence 
deviations are anticipated to be minimal, at the order of a few percent, 
they have the potential to compromise both the accuracy of treatment 
planning calculations and the overall clinical eQicacy of the delivered 
dose. Further investigation is required to quantify the magnitude of 
these eQects and to develop, if necessary, mitigation strategies to 
ensure consistent and predictable dose delivery. This may require the 
implementation of advanced techniques for the characterization of the 
beam spectrum and magnetic field profile, in conjunction with the 
refinement of treatment planning algorithms to account for these nonideal conditions. The objective is to achieve the highest level of 
dosimetric accuracy and treatment eQicacy, thus ensuring optimal 
patient outcomes.

ANSWER/REBUTTAL: The authors are in complete agreement with the Reviewer #3 
comment. Simulation outcomes would be reasonably representative of experimental 
data only if the simulation tool is properly and accurately implemented as well as if 
the simulation approach properly allows for considering realistic (non-idealized) 
conditions. In this regard, the capacity and the flexibility during the development and 
implementation of the simulation tools appear as key issues.

21. The research didn't fully integrate treatment planning and optimization 
algorithms, which are crucial for ensuring clinical eQectiveness. The 
diQiculties of individualized dosimetry and delivery optimum dose 
strategies are crucial for minimizing oQ-target eQects and maximizing 
targeted dose deposition within the clinical treatment paradigm. This 
represents a significant limitation, impeding a comprehensive 
assessment of the eQects under realistic clinical conditions. Further 
studies are required to incorporate these critical elements so as to 
evaluate the clinical translatability of the observed eQects.

ANSWER/REBUTTAL: The authors thank the Reviewer #3 for emphasizing the 
limitations arising from the lack of a method/tool for C ONVERAY treatment 
planning/optimization. Such an issue represents, in fact, one of the main challenges 
to be overcome during the next CONVERAY project phases. Moreover, as mentioned in 
previous items of this list, such a limitation might question the reliability of reporting 
in the current project stage about patient-specific dosimetry performance 
comparisons against existing radiotherapy technologies. Thereby, suggestions by the 
Editor and the Reviewers for re-organizing the manuscript as a proof-of-concept 
appeared as very valuable and attractive option that has driven the entire manuscript 
revision. 

22.In my view, the analysis and contextualization of the present findings 
within the existing body of literature should be some strengthen. The 
discussion section currently lacks suQicient depth in comparing the 
results with relevant peer research, limiting the study's contribution to 
the field. Because this study is a research article rather than a technical 
report, it would be advantageous to expand the discussion section.

ANSWER/REBUTTAL: The authors appreciate the Reviewer #3 comment. The revised 
manuscript proposes a completely modified and extended approach for the results’ 
analysis & discussions. 

23.As a scientific reader, I should find response to some of my questions as 
a reader of this article.
Given the manuscript's intended format as a research article, a more 
comprehensive discussion looks crucial. This necessitates a more 
thorough comparison with established literature and interpretation of 
the findings, exploring potential implications, limitations, and avenues 
for future research. Expanding this section could elevate the 
manuscript's scientific rigor and a more impactful contribution to the 
broader scholarly. Specifically, the discussion should address: How do 
the findings agreements or contradict previous works? (e.g., 
methodological diQerences, statical variations)? How do the results 
inform with existing MC frameworks? Do authors suggest new MC 
directions? How might these limitations aQect the interpretation of the 
findings? Table presentation could be beneficial for interpretation of 
some results? So, the results lack statistical analysis and some of the 
results should be tabulated.

ANSWER/REBUTTAL: The authors thank the Reviewer #3 for this comment. A new 
subsection has been added to the revised manuscript aimed at providing detailed 
descriptions of the relevant topics and features involved during the design, 
development, and implementation of the MC-based simulations. Furthermore, a 
specific subsection has been included to address the relevant issues involved in the 
uncertainty/accuracy performance. Please, refer to the subsection 2.3 Uncertainties 
of the CONVERAY Monte Carlo modelling

Reviewer 4 Report

Comments and Suggestions for Authors

The idea is innovative and interesting. The computational study presented in the manuscript is very comprehensive. Here are a few comments for improvement.

1.    The tense in the text should be past tense in general. Do not use “was” instead of “has been” etc.

2.    Figure 1 is the same as the figure in the authors’ previous publication, Ref.[4].

3.    Figure 2 shows the dose profile along the Z-axis. If so, I am surprised to see such a sharp peak. Explain how you change the potion of the peak in a patient. Differing from proton therapy, I assume you move the patient's body. It may not be easy to vary the electron beam energy.

4.    You should add labels in Figure 3 so that readers can see the components easier than reading the text.

5.    Figure 4 is not helpful. What does the color of the surface indicate?

6.    Indicate the X-Y-Z coordinate system in Figure 1 to understand Figures 8 and 9.

7.    Add a narrative description of the data in Figures 5 to 19 in the main text.

8.    The second paragraph on page 12 should be a new subsection 3.3.3 since the following results are for rotating gantry cases.

Comments on the Quality of English Language

1.    The tense in the text should be past tense in general. Do not use “was” instead of “has been” etc.

Author Response

The idea is innovative and interesting. The computational study 
presented in the manuscript is very comprehensive. Here are a few 
comments for improvement.

ANSWER/REBUTTAL: The authors would like to express honest and greatest thanks to 
the Reviewer #4 for the time and eQorts invested in reviewing the manuscript along 
with corresponding very positive feedback and valuable insights to improve it

25. The tense in the text should be past tense in general. Do not use “was” 
instead of “has been” etc.

ANSWER/REBUTTAL: The authors are grateful for this suggestion that has been 
accounted for and implemented.

26. Figure 1 is the same as the figure in the authors’ previous publication, 
Ref.[4].

ANSWER/REBUTTAL: The authors are very sorry for this unvoluntary inadvertent. The 
Figure 2 has been properly re-elaborated.

27. Figure 2 shows the dose profile along the Z-axis. If so, I am surprised to 
see such a sharp peak. Explain how you change the potion of the peak 
in a patient. DiQering from proton therapy, I assume you move the 
patient's body. It may not be easy to vary the electron beam energy.

ANSWER/REBUTTAL: The authors appreciate the Reviewer #4 
comment. Actually, the CONVERAY design relays -mainly- on the flux 
concentration by means of merely geometric reasons. Producing a 
“wide” photon beam (i.e. producing a photon beam by impact of 
electron onto a large area) allows to further collect such a high flux. The 
key issue is to collect photons travelling to the same destination (focal 
spot). If possible, the primary component of the photon flux shall 
achieve the desired flux concentration around the focal spot. In the 
presence of materials, interactions need to be included along with 
photon transport. Expression (1) [please refer to reference [4]] 
represents a simplified description of both phenomena that together 
determine -as a first approximation- the overall trend of the CONVERAY 
system.

28.You should add labels in Figure 3 so that readers can see the 
components easier than reading the text.

ANSWER/REBUTTAL: The authors thank the Reviewer #4 for pointing out this 
drawback that has been solved accordingly.

29. Figure 4 is not helpful. What does the color of the surface indicate?

ANSWER/REBUTTAL: The authors appreciate the Reviewer #4 suggestion. In fact, the 
authors agree that the Figure 4 original version may lack clarity, particularly in 
explaining the significance of the colors used to represent the magnetic field intensity.
In this regard, please consider a brief description to outline the intended approach:
• Clarification of the Color Representation:
The colors in Figure 4 represent the intensity of the magnetic field in the central plane 
of the deflection magnet, with a gradient indicating variations in field strength across 
the plane. To make the figure more informative, we will include a clear color bar with 
corresponding magnetic field strength values (in Tesla).
• Main purposes of Figure 4:
The figure is included to demonstrate the actual spatial distribution of the magnetic
field for guiding the electron beam through the CONVERAY system. authors have
clarified this issue by modifying the text citing the Figure 4, as follows: “ … For 
instance, Figure 4 reports the magnetic field central plane for one of the deflection 
magnets by illustrating the magnetic field strength in the central plane of the first deflection magnet, using corresponding experimental data that further serve as input for 
an accurate simulation of the electron beam along the intended path. Therefore, the 
uniformity of the magnetic field, as depicted in Figure 4, is critical for ensuring precise 
trajectory control of the electron beam within the CONVERAY system. This uniformity 
minimizes beam divergence and energy loss, which are essential for achieving the 
high fluence and dose conformity required for the system's performance.…" (L181-
L188)

30.Indicate the X-Y-Z coordinate system in Figure 1 to understand Figures 8 
and 9.

ANSWER/REBUTTAL: The authors are grateful to the Reviewer #4 for pointing out this 
issue. It has been addressed as required.

31.Add a narrative description of the data in Figures 5 to 19 in the main text.

ANSWER/REBUTTAL: The authors thank the Reviewer #4 for this suggestion. The 
revised version of the manuscript include general narrative information about the 
figures, while some specific features have been also added in the Discussion section.

32. The second paragraph on page 12 should be a new subsection 3.3.3 since the following results are for rotating gantry cases.

ANSWER/REBUTTAL: The authors appreciate the Reviewer #4 comment. In fact, this
unvoluntary mistake has been solved in the revised manuscript.

33.Comments on the Quality of English Language.
The tense in the text should be past tense in general. Do not use “was” 
instead of “has been” etc.

ANSWER/REBUTTAL: The authors thank the Reviewer #4 suggestion. It has been 
accounted for accordingly

Round 2

Reviewer 1 Report

Comments and Suggestions for Authors

The authors have successfully shifted the focus of the manuscript from "clinical suitability" to a proof-of-concept for the MC methods. Moreover, overall clarity and quality of the text have substantially improved. Over-statements, over-enthusiastic claims and vague language have been removed. I have no further comments.

Author Response

COMMENT: The authors have successfully shifted the focus of the manuscript from "clinical suitability" to a proof-of-concept for the MC methods. Moreover, overall clarity and quality of the text have substantially improved. Over-statements, over-enthusiastic claims and vague language have been removed. I have no further comments.

ANSWER: The comments provided by Reviewer #1 during the R1 and R2 rounds are greatly appreciated. The positive assessment of the initial manuscript structure as a proof-of-concept is acknowledged with gratitude. The suggestions offered by Reviewer #1, particularly about avoiding over-statements, have had a significant impact on the manuscript's improvement. The revisions made in response to these comments have substantially enhanced the current version.

Reviewer 3 Report

Comments and Suggestions for Authors

In this revised version, it is evident that the authors have addressed the majority of the criticisms proposed, with significant alterations having been made to the original work. It is evident that this iteration of the article exhibits a heightened scientific rigor.

while the manuscript is substantially improved, a few points require further attention. In order to increase comprehensibility, the explanation of the color scale in Figure 4 should be stated under the figure, even in a sentence. In addition, such situations should be checked for all figures and revised if any. The scale of Figure 6 cannot be fully seen in the version uploaded to the system. This should be taken into consideration before publishing.

Addressing the remaining easy minor points outlined above would further enhance and justify publication. I recommend acceptance of the paper. It is not necessary to re-examine. BW

Author Response

COMMENT: 

In this revised version, it is evident that the authors have addressed the majority of the criticisms proposed, with significant alterations having been made to the original work. It is evident that this iteration of the article exhibits a heightened scientific rigor.

while the manuscript is substantially improved, a few points require further attention. In order to increase comprehensibility, the explanation of the color scale in Figure 4 should be stated under the figure, even in a sentence. In addition, such situations should be checked for all figures and revised if any. The scale of Figure 6 cannot be fully seen in the version uploaded to the system. This should be taken into consideration before publishing.

Addressing the remaining easy minor points outlined above would further enhance and justify publication. I recommend acceptance of the paper. It is not necessary to re-examine. BW

ANSWER:

The comments and recommendations provided on the revised manuscript are greatly appreciated. Authors appreciate that the Reviewer #3 recognized that the majority of the criticisms raised had been successfully addressed during the R1 round, and the manuscript has significantly improved in terms of scientific rigor.

In response to the specific comments, the following modifications have been performed:

  • A clear explanation of the color scale has been added to Figure 4, as suggested. The revised caption of Figure 4 states as: " Magnetic field strength in the central plane (equidistant from the neodymium magnets) for the first (the first one chronologically attained by the incident electron beam) deflection magnet. Un-certainties in the experimental data are less than 2 %. The beam propagation direction (optical axis), denotes as +z, enters the magnetic field according to the required deflection (tilt) angle. Surface colors indicate the magnetic field strength (from 0.1 T in blue to 1 T in yellow)." The text in blue has been added to clarify the issue. It is worth mentioning that Figure 4 has been re-elaborate to meet the query from the Reviewer #3, including the corresponding color-bar.

  • All figures have been carefully reviewed to ensure that the explanations of the color scales are clear and precise. Some minor, but useful, clarifications have been added to Figures 5, 6, and 14.

  • The scale of Figure 6 has been verified to be fully visible in the R2 revised version of the manuscript.

The issues raised have been thoroughly addressed, thus the authors expect that the R2 revised manuscript version may properly attain the Reviewer #3 queries, whose recommendations have been greatly appreciated.

Reviewer 4 Report

Comments and Suggestions for Authors

Thank you for the revisions and answering my questions.

Author Response

COMMENT: 

Comments and Suggestions for Authors: Thank you for the revisions and answering my questions

ANSWER:

The authors are deeply grateful to Reviewer #4 for providing meticulous evaluation of the manuscript during both the R1 and R2 rounds. The Reviewer #4’s positive feedback and insightful comments have been instrumental in refining the manuscript, and their investment of time and effort is sincerely appreciated.